# Review of Molecular Dynamics Simulation of Bimetallic Interfacial Behavior

**DOI:** 10.3390/ma18133048

**Published:** 2025-06-26

**Authors:** Xiaoqiong Wang, Yuejia Wang, Guangyu Li, Wenming Jiang, Jun Wang, Xing Kang, Qiantong Zeng, Shan Yao, Pingkun Yao

**Affiliations:** 1School of Materials Science and Engineering, Dalian University of Technology, Dalian 116024, China; xqw@mail.dlut.edu.cn (X.W.); dgwangyuejia@mail.dlut.edu.cn (Y.W.); xingkangx@163.com (X.K.); zqt_185@163.com (Q.Z.); yaoshan@dlut.edu.cn (S.Y.); ypingkun@dlut.edu.cn (P.Y.); 2Ningbo Institute of Dalian University of Technology, Dalian University of Technology, Ningbo 315000, China; 3State Key Laboratory of Materials Processing and Die & Mould Technology, School of Materials Science and Engineering, Huazhong University of Science and Technology, Wuhan 430074, China; 4State Key Laboratory of Solidification Processing, Northwestern Polytechnical University, Xi’an 710072, China; nwpuwj@nwpu.edu.cn

**Keywords:** bimetal, interfacial behavior, mechanical properties, molecular dynamics simulation, research progress, prospect

## Abstract

Bimetals have broad application prospects in many fields due to the combination of the performance characteristics of the two materials, but weak interface bonding limits their promotion and application. Therefore, studying the interfacial behavior to achieve bimetallic strengthening is the focus of this field. However, it is often difficult or costly to visually observe the interfacial behavior using traditional experimental methods. Molecular dynamics (MD) is an advanced microscopic simulation method that can conveniently, rapidly, accurately and intuitively study the diffusion and mechanical behavior at the bimetallic interfaces, providing a powerful tool and theoretical guidance to reveal the nature of interfacial bonding and the strengthening mechanism. This paper summarizes the research progress on molecular dynamics in the bimetallic formation process and mechanical behavior, including Al/Cu, Al/Mg, Al/Ni, Al/Ti, Al/Fe, Cu/Ni, and Fe/Cu. In addition, the future development direction is outlined to provide theoretical basis and experimental guidance for further exploring the formation process and performance enhancement of the bimetallic interfaces.

## 1. Introduction

In comparison to traditional single-material systems, bimetallic materials effectively integrate the advantageous properties of two distinct materials, thereby better fulfilling performance requirements in extreme working environments [1,2,3]. This characteristic renders them highly promising for applications in aerospace, automotive, and defense sectors [4,5,6]. However, the current limitation lies in the low interface strength of bimetals, which significantly hinders their application. Consequently, investigating the interfacial behavior of bimetals is crucial to enhancing their performance.

Currently, two primary methodologies are employed to investigate the behavior of bimetallic interfaces: experimental techniques and computational modeling [7,8]. The microstructure of these interfaces is typically examined and analyzed using scanning electron microscopy (SEM), transmission electron microscopy (TEM), X-ray diffraction (XRD), and energy dispersive spectroscopy (EDS) [9,10,11,12,13]. However, experimental approaches are constrained by several limitations, including the difficulty of real-time observation of dynamic interfacial changes, extended time requirements, high costs, the stochastic nature of experiments, and the necessity for repeated verification [14,15,16]. These constraints pose significant challenges for experimental studies on bimetallic interfacial behavior.

As a highly effective complementary tool to traditional theoretical and experimental methods, computer simulation offers distinct advantages [17]. Predictive results consistent with experimental data can be obtained through well-conducted simulations, while dynamic physicochemical processes currently difficult to observe experimentally can also be simulated [18]. Furthermore, simulation methods are not restricted by experimental conditions, allowing for easy modification of experimental parameters and boundary conditions to assess the impact of various factors on performance [19].

Computational simulations can be categorized into macroscopic, mesoscopic, and microscopic dimensions [20,21]. Macroscopic simulations struggle to accurately depict the microstructure and interactions at the interface, rendering them unsuitable for studying bimetallic interface lines [22]. Mesoscopic simulations, primarily utilizing the phase field method and cellular automata, focus on mesoscopic scale phenomena such as the size effect of solder joints and phase component analysis [23], yet they fall short of revealing atomic-level interface lines [24,25]. In contrast, microscopic simulations can elucidate the microscopic details and interaction mechanisms at the interface, making them a crucial tool for studying bimetallic interfacial behavior. In recent years, artificial intelligence-driven MD simulations have significantly enhanced simulation accuracy and efficiency by integrating machine learning potential functions with high-performance computing. This advancement has provided a new paradigm for the dynamic analysis of complex interfacial behaviors [26,27,28].

Among microscopic simulations, several methods are available for investigating bimetallic interfacial behavior, including first-principles calculations based on density functional theory (DFT) [23,29], Monte Carlo simulations grounded in statistical physics [30], and molecular dynamics simulations based on classical Newtonian mechanics [31,32]. First-principle calculations employ quantum mechanics to solve the Schrödinger equation, thereby determining the electronic structure, formation energy, state density, thermal conductivity, electrical conductivity, thermal expansion coefficient, and elastic constants, among other properties [33,34]. Although these simulations yield highly accurate results, they are limited by small system sizes and are time consuming. The Monte Carlo method excels in handling large-scale systems and calculating thermodynamic properties when simulating bimetallic interfacial behavior; however, it has limitations in terms of dynamic behavior and convergence. While molecular dynamics simulations may not match the accuracy of first-principles calculations, they can model and simulate systems ranging from a few particles to millions or billions of particles, requiring less time than first-principles methods and capturing atomic dynamic behavior. Consequently, molecular dynamics simulations have significant advantages in studying bimetallic interfacial behavior and have become the primary means for simulating such behavior. However, there are currently no review articles on the molecular dynamics simulation of bimetallic interfacial behavior.

To study the interfacial behavior of bimetallic systems using molecular dynamics, the process begins by constructing an initial interface model based on the selected metals and their crystal structures, while setting appropriate periodic boundary conditions. This was followed by the relaxation of atomic positions through energy minimization. The core step involves selecting and validating high-precision atomic potential functions such as the EAM or MEAM. Subsequently, the simulation parameters, including the timestep, ensemble, and temperature or pressure control, were configured. After ensuring system stability through sufficient equilibration, atomic trajectories and thermodynamic data were collected. In the final stage, atomic configurations were visualized, and the interfacial structure, energy evolution, and dynamic processes were analyzed using methods such as the Centrosymmetry Parameter and Mean Squared Displacement [35,36,37,38].

This study examines recent advancements in molecular dynamics research concerning the behavior of bimetallic interfaces. This study concentrates on two primary aspects: the process of interface formation and the mechanical behavior of the interface. Regarding the interface formation process, the analysis explores the effects of internal factors, such as lattice mismatch, interface structure, and type of intermetallic compounds, alongside external factors, such as temperature and applied load state, on the dynamic evolution of the interface at the atomic level. Concerning the mechanical behavior of the interface, this study systematically analyzes dislocation evolution and deformation characteristics during tensile and compressive processes and further discusses in detail the regulatory effects of various internal and external factors on mechanical behavior, as represented in Figure 1. Additionally, this article anticipates future research directions in this domain, emphasizing the development of machine-learning atomic potentials with enhanced accuracy and efficiency, the advancement of cross-scale simulation methods integrating multiple scales, and the exploration of innovative applications of artificial intelligence in accelerating interface simulations, predicting performance metrics, and optimizing interfacial structures. This study aims to provide theoretical guidance for a more profound understanding of the behavioral mechanisms and performance optimization of the bimetallic interface.

## 2. Molecular Dynamics Simulation of Al/Cu Bimetallic Interfacial Behavior 

### 2.1. Interface Formation Process of Al/Cu Bimetallic Systems

The formation process of bimetallic interfaces encompasses atomic diffusion, intermetallic compound formation, and the melting and solidification of metals. This process is influenced by both material internal factors, such as interface orientation and crystal structure, and external factors, including preparation process, temperature, pressure, and time. The following section summarizes relevant molecular dynamics simulation research on the formation process of the Al/Cu bimetallic interface, considering both internal and external factors.

Internal Influencing Factors

Yang JW et al. [39] employed MD to investigate the diffusion process of Al/Cu joints with typical large-angle grain boundaries (LAGBs). The findings indicate that LAGBs significantly enhance the diffusion of Cu atoms into the Al matrix, achieving a maximum diffusion distance of 2.6 nm. In contrast, the diffusion of Al atoms into the Cu matrix is comparatively weaker, with a maximum penetration distance of 2.2 nm. Furthermore, the atomic ratio of Cu to Al at the interface approached 1:2, leading to the formation of Al_2_Cu intermetallic compounds.

Zhang JP et al. [40] examined the diffusion and solidification process of the CuGB/Al solid–liquid interface at different Cu grain boundary phase angles through molecular dynamics simulation. The simulation results reveal a grain boundary heredity in the system: when a vertical grain boundary exists in solid Cu, the Cu atoms near the grain boundary diffuse into the liquid Al region and form a horizontal grain boundary of Al/Cu upon cooling. Furthermore, initial grain boundary phase angle significantly influence the crystal structure post-solidification. Specifically, the proportion of the FCC phase decreases with an increase in the Cu grain boundary phase angle, while the proportion of the HCP phase after solidification exhibits an opposite trend, as depicted in Figure 2.

Chen SD et al. [41] investigated the influence of surface roughness on the diffusion processes of Cu and Al through MD simulations. The research results show that when there is roughness at the interface, the diffusion connection process is divided into three stages. In the first stage, surfaces deform under stress, resulting in increased contact areas, as seen in Figure 3b. In the second stage, the increase in temperature prompts the deformation of the softer aluminum surface and eliminates the interface gap. This process culminated in the final states presented in Figure 3(d1–d3). Figure 3(d1) clearly shows the complete flattening of the rough aluminum surface at 700 K, achieving a gapless contact. Figure 3(d2) confirms that the Al atoms completely filled the Cu surface depressions, forming a continuous interface. Figure 3(d3) shows that the dual rough surfaces eliminate interfacial gaps through Al-dominated deformation, with the bonded interface morphology being closer to the initial roughness of Cu. The third stage corresponds to constant-temperature diffusion, during which atomic interdiffusion across the interface becomes markedly enhanced when the temperature exceeds 600 K. Regardless of the roughness distribution, the deformation of aluminum is always the main mechanism for eliminating the interfacial gap, and the interface morphology after the connection is closer to the roughness of the initial copper, as shown in Figure 3.

External Influencing Factors

Numerous researchers have conducted extensive investigations into the impact of temperature on the Cu/Al bimetallic interface formation process. However, owing to variations in the experimental conditions and other factors, the specific simulation results obtained from different studies exhibit some discrepancies.

Han XJ et al. [42] simulated the effect of varying holding temperatures on atomic diffusion at the Al/Cu bimetallic interface. They discovered that there was negligible diffusion between the interface atoms below 750 K, while significant diffusion occurred above 750 K. A temperature of 800 K is identified as the optimal holding temperature of simulated interface diffusion. Cui YF et al. [43] conducted a study utilizing molecular dynamics methods to examine the effects of various holding temperatures on the diffusion characteristics at the Cu/Al_2_Cu interface and the Al/Al_2_Cu interface within the Cu/Al_2_Cu/Al system. The results show that at the Cu/Al_2_Cu interface, the Cu atoms can be deeply diffused into the interior of Al_2_Cu, while the Al_2_Cu can only diffuse towards the Cu side near the interface at 933 K or above. The Cu layer can maintain a high degree of order, but at 933 K and above, the structural order of the Cu layer is gradually destroyed. At the Al/Al_2_Cu interface, the phenomenon of mutual diffusion between Al atoms and Al_2_Cu is very obvious, and the ordered structure of the Al layer is destroyed during the diffusion process, showing a chaotic state. Mao AX et al. [44] further investigated the temperature dependence of atomic interdiffusion perpendicular to the Cu(110)/Al solid–liquid interface. The results demonstrate that both the diffusion coefficients of Cu/Al atoms and the diffusion depth of Al atoms obey the Arrhenius equation with temperature. In contrast, the diffusion depth of Cu atoms exhibits a linear dependence on temperature, as evidenced by Figure 4. From the Arrhenius formula lnD = lnD_0_ − Q/(RT), the diffusion activation energy Q of atoms can be obtained. The calculated diffusion activation energies Q of Cu and Al atoms are 130.7 kJ/mol and 18.8 kJ/mol, respectively. Multiple studies have quantified the diffusion coefficients of Cu and Al across temperatures, including work by Mao AX et al. [44] and Qian XF et al. [45], as shown in Table 1. Li JR et al. [46] noted that below 750 K, copper and aluminum maintain a face-centered cubic (FCC) phase, with copper atoms diffusing into the Al-FCC phase. When the temperature exceeded 750 K, a substantial portion of aluminum transformed into a disordered structure, allowing copper atoms to diffuse into the disordered aluminum phase.

In addition to temperature, time factors such as pouring time, holding time, and diffusion time also significantly affect the diffusion behavior of Al/Cu bimetals, as presented in Figure 5. Qian XF et al. [45] investigated the effect of different pouring times on Al/Cu bimetallic interface diffusion using MD simulations. Their findings indicate that as pouring time increases, the inter-diffusion of Cu and Al atoms in the interface region rises, leading to an increase in the transition layer thickness, as indicated in Figure 5a. Han XJ et al. [42] observed that with increasing holding time, the transition layer thickness initially increased but subsequently remained relatively constant, as displayed in Figure 5b. Mao AX et al. [44] identified a parabolic relationship between diffusion layer thickness and diffusion time, which aligns well with experimental observations, presented in Figure 5c.

Furthermore, pressure plays a crucial role in determining the diffusion rate and path of Al/Cu bimetallic interfacial atoms, representing one of the key factors influencing interfacial diffusion behavior, as presented in Figure 6. Li QH et al. [47] examined the impact of pressure on the interface diffusion behavior of Cu/Al bimetal friction stir welded joints, revealing that external pressure had minimal influence on diffusion thickness, as depicted in Figure 6a. Mao AQ et al. [44] employed a molecular dynamics method to simulate the impact of pressure on atomic diffusion behavior at the Al/Cu solid–liquid interface. It was found that with an increase in external pressure, the diffusion depths of both Cu and Al atoms decreased, indicating that high pressure inhibited the diffusion process. When the pressure ranged from 1 to 5 bar and the temperature was 950 K, the predominantly formed intermetallic compound was θ-Al_2_Cu, as shown in Figure 6b. These findings suggest that the influence of pressure on interfacial diffusion varies across different processes.

Other factors, such as collision velocity and reduction rate, also affect the diffusion behavior of the Al/Cu bimetallic interface.

Chen SY et al. [48] studied the atomic diffusion behavior in the process of Al/Cu explosive welding and found that atomic diffusion mainly occurred in the unloading stage of the welding process. The diffusion coefficient depends on the collision velocity, and the higher the velocity, the greater the coefficient. When there is no transverse velocity, the diffusion coefficient is proportional to the longitudinal velocity. When the longitudinal velocity is fixed, the diffusion coefficient is proportional to the square of the transverse velocity.

Li JR et al. [49] investigated the atomic diffusion behavior at the Al/Cu interface during rolling processes through MD simulations. With an increasing rolling reduction ratio from 0 to 0.6, the interfacial interdiffusion between Cu and Al atoms leads to progressive formation of IMCs, including AlCu, Al_2_Cu, and Al_4_Cu_9_. The interfacial structure evolves from an initial biphasic Al/Cu state at reduction ratios below 0.3 to a triphasic Cu-IMCs-Al coexistence configuration at reduction ratios of 0.4 or higher, as highlighted in Figure 7. Furthermore, the amount of IMCs formed by rolling is the largest when the size ratio of Cu to Al is the lattice constant ratio.

### 2.2. Mechanical Behavior of Al/Cu Bimetallic Interfaces

Currently, research on the deformation behavior of bimetallic interfaces primarily focuses on uniaxial tension and uniaxial compression. Influencing factors include internal aspects, such as the formation of intermetallic compounds, material defects, and crystal orientation, as well as external factors such as temperature, pressure, and strain rate.

#### 2.2.1. Tensile Behavior

Internal Influencing Factors

Bian XQ et al. [50,51] investigated the impact of the presence or absence of the interfacial compound Al_2_Cu and the layer thickness on the deformation behavior of nano-polycrystalline Al/Al_2_Cu/Cu layered composites using MD simulations. Their findings indicate that the inclusion of Al_2_Cu as an interface layer enhances the strength and toughness of the Cu/Al_2_Cu/Al model compared to the nanocrystalline Al/Cu model without Al_2_Cu, as illustrated in Figure 8. The ultimate strength and fracture mode of the nano-polycrystalline Al/Al_2_Cu/Cu lamellar composite do not exhibit a monotonic relationship with the thickness d of the single crystal Al_2_Cu composition layer. The ultimate strength peaks at d = 2.44 nm, while the toughness reaches its optimal value at d = 4.88 nm. The interface layer Al_2_Cu plays a decisive role in effectively preventing microcrack propagation in the Cu/Al_2_Cu/Al model and promoting the co-deformation between the matrix Cu and Al.

Chen Y et al. [52] examined the micromechanical behavior of Al/Al_2_Cu/Cu bimetals with characteristic crystal orientations during uniaxial tensile deformation using MD simulations. The results demonstrate that dislocations nucleate at the Cu/Al_2_Cu heterogeneous interface and propagate along the {111} crystal plane towards the copper layer under tensile loading. The deformation mechanism is characterized by confined slip within the layer. As the load increases, stress concentration at the Al_2_Cu/Al interface ultimately leads to the fracture of the composite, as presented in Figure 9.

Wang AQ et al. [53] investigated the effects of solid solution and vacancies on the tensile deformation behavior of Cu(Al)/Al_2_Cu/(Cu)Al gradient heterostructures by MD simulations. The results reveal that the layered heterostructure containing Al-20% Cu exhibits the best strength–plasticity synergistic effect. The gradient Cu solute improves the synergistic tensile deformation behavior among the components, retards microcrack initiation, as seen in Figure 10(b-2), and has a higher ultimate strength and fracture strain than a single-layer solid solution. Additionally, the presence of vacancies concentrates strain in the polycrystalline Cu adjacent to the interface, as seen in Figure 10(b-4), weakening strain delocalization in polycrystalline Al and causing the material to fail at low tensile strains, as presented in Figure 10.

The mechanical properties of Al/Cu bimetals are closely associated with interfacial orientation. Pang WW et al. [54] explored the influence of different interface orientations on the deformation behavior of Al/Cu bimetals during tensile processes. The models with varying interface orientations are presented in Table 2. It was observed that the tensile stress and strain in models A and B are higher than those in models C and D. In models with different interfacial orientations, distinct forms of voids emerge during the stretching process. In models A and B, spherical and disk-shaped vacancies initially appear near the interface, with the vacancies primarily expanding within and around the aluminum layer, respectively. In contrast, for models C and D, strip voids first manifest along the intersection line between the slip dislocation and the interface, with the voids predominantly expanding along the interface in the [110] and [$\overset{-}{1}10$] directions, respectively, as depicted in Figure 11.

External Influencing Factors

The influence of various preparation processes and their parameters on the interfacial mechanical behavior of Al/Cu bimetal during tensile deformation was examined through MD simulations.

Wang XL et al. [55] investigated the nanostructure evolution and mechanical properties of the Al/Cu interface using a multiscale approach, focusing on the effect of the annealing temperature on the interfacial strength. The study identified the formation of Al_2_Cu (Al-side) and Al_4_Cu_9_ (Cu-side) intermetallic compounds in the interfacial region using K-means cluster analysis. The simulation results show that the tensile and shear strengths of the Al_2_Cu-Al interface are lower than those of the Cu-Al_4_Cu_9_ interface, and the failures start from the Al side; the tensile strength of the interface on the Al side increases and then decreases, while the shear strength decreases and then increases with an increase in the annealing temperature. The mechanical properties of the Cu-side interface were less affected by annealing temperature. At an annealing temperature of 573 K, the Al-Cu interface exhibited the best overall mechanical properties.

Wang MD et al. [56] explored the effect of varying strain rates on the interfacial tensile strength of Al/Cu-Zr composites. Their results reveal that while the plastic deformation mechanisms at different strain rates are similar, the fracture modes differ significantly. Specifically, ductile fracture occurs at strain rates of 0.01 ps^−1^ and 0.001 ps^−1^, whereas brittle fracture is observed at a strain rate of 0.0001 ps^−1^, as depicted in Figure 12. Jin YH et al. [57] further analyzed the interfacial mechanical properties under different strain rates. They demonstrated that the tensile strength and yield strength at the interface increase with rising tensile strain rate, and the maximum interfacial tensile strength of 3.19 GPa is achieved at a tensile strain rate of 1 × 10^10^ s^−1^.

Pang WW et al. [58] investigated the impact of loading axis orientation on the deformation behavior of Al/Cu bimetals through MD simulations. Their results indicate that the yield strength is lower when the load is applied parallel to the interface (x or y axis) compared to perpendicular loading (z axis), whereas the opposite trend is observed for tensile strength, as shown in Figure 13. The deformation mechanisms along the x-, y-, and z-axes involve partial dislocations and deformation twins, extended total dislocations, and partial dislocations, respectively.

The integration of alloying elements can markedly improve the bonding strength at the Al/Cu interface. Wang MD et al. [56] explored the impact of varying Zr content on the interfacial tensile strength of Cu-Zr/Al composites. Their results suggested that with an increase in the Zr content, the tensile strength of the Al/Cu bimetal initially increased and then decreased, reaching its maximum at a Zr content of 0.33 wt%.

The mechanical properties of bimetals, such as strength, hardness, ductility, and fracture toughness, can be influenced by the interaction of external electric fields (EEFs) with nanoparticles in the metal matrix. Gao XB et al. [59] assessed the effects of various EEF values on the mechanical properties of Al/Cu/Al three-layer nanocomposites (TLNCs) using the MD modeling method. When EEF was not applied, the ultimate tensile strength (UTS) and the Young’s modulus (YM) of the model were 2.800 GPa and 21.286 GPa, respectively. By raising the EFF value, the physical and mechanical stability strength of the samples were reduced. Specifically, when the EEF reached 0.05 V/Å, the UTS and YM decreased to 2.587 GPa and 20.19 GPa, as outlined in Figure 14.

#### 2.2.2. Compression Behavior

Internal Influencing Factors

Liu XP et al. [60] investigated the deformation compatibility of Al/Cu bimetals with various interface structures under compression using MD simulations; the four model structures are shown in Figure 15. The findings reveal that, compared to homogeneous structures, heterogeneous Al/Cu nanocomposites with a ($\overset{-}{1}\overset{-}{1}1$) interface exhibit superior deformation compatibility while the homogeneous counterpart does not. Heterostructures with a (001) interfacial structures display pronounced non-coordinated deformation among their components. A comparison of the strength data from the four models reveals that the models with ($\overset{-}{1}\overset{-}{1}1$) interfaces exhibit overall higher strength than those with (001) interfaces. Additionally, the strength of homogeneous structures is slightly greater than that of heterogeneous structures under the same interface conditions, as illustrated in Table 3.

External Influencing Factors

Yin FX et al. [61] explored the interfacial evolution characteristics and deformation mechanisms of Al/Cu bimetal during compression at varying strain rates through MD simulations. It was observed that both yield strength and ductility slightly enhance with increasing strain rate, accompanied by progressive interface roughening. Notably, the strain rate does not alter the evolution way of dislocation networks. Under all tested strain rates, the stress–strain curves consistently display two characteristic yield points. The first yield point correlates with the decomposition of perfect misfit dislocations on the interface and the propagation of partial dislocations inside the Al layer, while the second yield point relates with the dislocation transmission from the Al layer into the Cu layer, as indicated in Figure 16.

## 3. Molecular Dynamics Simulation of Al/Mg Bimetallic Interfacial Behavior 

### 3.1. Interface Formation Process of Al/Mg Bimetallic Systems

Similar to the Al/Cu bimetal, the interface formation process in the Al/Mg bimetal is influenced by various factors, primarily dominated by external factors such as collision rate and collision angle. In contrast, research on the influence of internal factors is relatively limited.

External Influencing Factors

The primary fabrication methods for Al/Mg bimetallic materials include rolling bonding, explosive bonding, extrusion bonding, and diffusion bonding. Under varying preparation processes, distinct process parameters exert differing influences on the interface formation mechanism [117]. Zhang TT et al. [62] investigated the atomic diffusion behavior at the bonding interface of Al/Mg composite plates during explosive welding using MD simulations. The findings indicate that the diffusion coefficient of Mg atoms is greater than that of Al atoms, with the disparity between Mg and Al atoms decreasing as the collision rate increases. The diffusion coefficient is influenced by both the collision rate and angle, with specific results presented in Figure 17. Additionally, a formula for the diffusion layer thickness was derived from the simulated data, and its reliability was confirmed through experimental validation.

### 3.2. Mechanical Behavior of Al/Mg Bimetallic Interfaces

#### 3.2.1. Tensile Behavior

Internal Influencing Factors

The intermetallic compounds at the bimetallic interface significantly affect the mechanical behavior of the interface. They play a crucial role in the tensile process by altering the bonding strength, toughness, and fatigue resistance of the interface, thereby influencing the overall mechanical properties of the material.

Li Y et al. [63] explored the tensile behavior of Mg-Mg_2_Al_3_-Al nanostructures (CS1) and Mg-Mg_17_Al_12_-Al (CS2) nanostructures through MD simulations, comparing them with three other nanostructures and substrates. The results reveal that the maximum stress of these two Mg-Al composites is substantially higher than that of pure amorphous MgxAly, although the stress and corresponding strain values are only approximately 60–70% of Mg. The fracture location of CS1 is within the magnesium block, while the final fracture position of CS2 is at the interface between Mg and Mg_17_Al_12_, as depicted in Figure 18.

Utilizing MD simulations, Li Z et al. [64] conducted a comprehensive investigation into the effects of various factors, such as grain size and intermetallic compound layer thickness, on the plastic deformation behavior of nanocrystalline Al/Mg bimetals. The results indicate that the effect of grain size on deformation behavior is dependent on the strain rate. At low strain rates, an increase in grain size is inversely proportional to yield stress, whereas at high strain rates, an increase in grain size is proportional to tensile stress. Furthermore, an optimal interlayer thickness enhances the strength of the composite, while an excessively thick interlayer reduces tensile strength due to a decrease in grain boundaries available to accommodate dislocation.

External Influencing Factors

Temperature is one of the crucial external factors that significantly influence the mechanical properties of Al/Mg bimetal. Lv XQ et al. [65] found that the yielding of nanolayers primarily originates from the Mg layer and is independent of the thermal diffusion temperature T1, while failure is predominantly caused by the Al layer. At lower temperatures, both tensile strength and corresponding strain slightly decrease with increasing thermal diffusion temperature. At elevated temperatures, the formation of coherent Al/Mg interfaces leads to a significant enhancement in tensile strength and ductility. The nanolayer in the case with T = 818 K exhibits the best comprehensive tensile properties.

Numerous scholars are engaged in the experimental investigation of high strain rate fracture in metals. Li Z et al. [64] utilized MD simulations to examine the impact of strain rate on the plastic deformation behavior of nanopolycrystalline Al/Mg bimetals. The findings indicate that an increase in strain rate results in changes to the grain size of the 6.29 nm Al/Mg layer, as well as enhancements in the bimetallic elastic modulus, tensile strength, and yield stress. Additionally, the strain rate influences the fracture behavior of the material, affecting crack extension and fracture morphology, as depicted in Figure 19. Pogorelko et al. [66] conducted high-speed uniaxial MD simulations on Al/Mg nanocomposites to assess their tensile strength under tensile conditions. At a strain rate of 10^9^ s^−1^ and a temperature of 300 K, voids were observed to form within the Mg-Al nanocomposites, initiating within the magnesium matrix and ultimately leading to fracture.

It has been suggested that the incorporation of interlayer coatings can serve a dual function: regulating the plastic deformation capacity of the two metals and controlling the growth of intermetallic compounds, thereby achieving interfacial regulation [118]. To enhance the bonding strength of Al/Mg bimetallic alloys, Xu YC et al. [67] introduced a FeCoNiCrCu high-entropy alloy coating at the interface of composite-cast Al/Mg bimetallics. MD simulations confirmed that the FeCoNiCrCuHEA coating effectively inhibited the diffusion between Al and Mg and the formation of Al-Mg intermetallic compounds, significantly reducing the interface thickness to 9.86% of the original. The shear strength increased from 30.62 MPa to 50.47 MPa, representing a 64.83% improvement, as illustrated in Figure 20.

Ultrasonic vibration effectively improves material microstructure uniformity and mechanical performance. Li XY et al. [68] studied the structural evolution of Al/Mg nanolayers under two conditions within the temperature range of 600–800 K: heating alone and heating with simultaneous application of ultrasonic vibration. They found that ultrasonic vibration can promote the formation and movement of dislocations, as well as atomic diffusion. In particular, the atomic displacement in the X-direction is nearly 1000 times greater than that without ultrasonic vibration. This causes the Al/Mg interface to change from a wavy shape to a plate-like shape and become more uniform, and the interface thickness increases. Ultrasonic vibration with large amplitudes (B ≥ 5 nm) and a low frequency (f = 5.7 GHz) tends to make the interface structure highly uniform, as illustrated in Figure 21. However, limited by the size of the nanolayers, the effect of amplitude on increasing the interface thickness is restricted.

#### 3.2.2. Compression Behavior

Internal Influencing Factors

The deformation mechanism of the Al/Mg nanomultilayer composite material with an FCC/HCP incoherent interface under a compressive load was systematically studied by Li Z et al. using an MD simulation method [69]. Atomic models with different layer thicknesses (2.5–80 nm) and slip angles (0–75°) were constructed. The size and orientation dependence of the mechanical properties of the material were revealed. A peak strength of 6 GPa was reached when the layer thickness was 26.7 nm. As the slip angle increased, the deformation mechanism changed from dislocation-dominated on the Al side to basal-plane-slip-dominated on the Mg side. The interface compatibility improved with an increase in the angle. Based on these findings, a dual-mode CLS model was proposed, as illustrated in Figure 22.

External Influencing Factors

In practical applications, the loading of interfaces is often complex, typically involving composite loading. Consequently, it is imperative to investigate the deformation of composite interfaces under multiaxial or composite loading conditions. Polyakova PV et al. [70] employed MD simulation methods to examine and analyze atomic motions at the Al/Mg interface under the combined influence of high pressure and shear strain. The findings indicated that under combined loading of 0.04 compressive strain and 0.4 shear strain, the Al/Mg system exhibits pronounced atomic mixing, predominantly occurring during the initial straining stage (a < 0.4). During this stage, the mixed zone demonstrates inferior mechanical strength, leading to interfacial fracture. With further strain accumulation, the compressive effect significantly suppresses atomic mixing behavior, as illustrated in Figure 23.

## 4. Molecular Dynamics Simulation of Al/Ni Bimetallic Interfacial Behavior 

### 4.1. Interface Formation Process of Al/Ni Bimetallic Systems

The substantial differences in the physical properties of aluminum and nickel, such as melting points and linear expansion coefficients, render the joining of these two metals particularly challenging. A comprehensive understanding of the interface formation process can mitigate adverse effects [119].

External Influencing Factors

Numerous scholars have extensively examined the effect of temperature on the formation of the Al/Ni interface through MD simulations, addressing key issues such as atomic diffusion behavior and the types of intermetallic compounds formed.

Firstly, concerning the effect of temperature on the diffusion behavior of the Al/Ni interface, Zhang CG et al. [71] conducted MD simulations of Al/Ni interfacial diffusion. The results demonstrate that when the temperature is below 700 K, there is no significant interfacial diffusion; however, when the temperature rises to 800 K, interfacial atoms begin to undergo interdiffusion. Furthermore, several studies [72,73,74] have indicated that the thickness of the diffusion layer increases with rising temperature, as depicted in Figure 24.

Secondly, the distinct interfacial reaction mechanisms directly determine the types of intermetallic compounds formed. This relationship has been thoroughly verified by many researchers through studies of different systems using MD simulations, among other methods. Smith GD et al. [75] performed MD simulations of the reaction process of the Al/Ni bimetallic system under isothermal conditions (900 K–1700 K). At lower temperatures, the interface initially forms the Al_3_Ni_2_ phase, which subsequently transforms into AlNi with the B2 structure; at higher temperatures, AlNi is directly formed at the interface. Politano O et al. [76,77] conducted a MD study of the layered Ni-Al-Ni system using the embedded atom potential method, with an initial heat treatment performed at a constant temperature of 600 K. Interdiffusion of Ni and Al at the interface was observed, leading to the spontaneous formation of the B2-NiAl phase within the Al layer. As time and temperature increase to 1000 K, the system may partially lose its B2-NiAl ordering in favor of L12-structured Ni_3_Al. The nucleation and growth mechanisms of the intermetallic phase were subsequently characterized. Kunwar A et al. [78] combined nanoscale MD simulations with mesoscopic phase field modeling to investigate the growth phenomena of Al_3_Ni_2_ at the Al/Ni interface at 1173.15 K. Their study revealed that various intermetallic compounds can form under specific reaction conditions and atomic diffusion states.

The temporal parameter exerts significant influence on the interfacial dynamics and structural evolution of Al/Ni interfaces, particularly with respect to intermetallic compound formation and diffusion-controlled growth mechanisms. Kolli S et al. [79] elucidated deformation mechanisms during the tensile and shear of pure Al, Ni, and Ni-Al interfacial systems. The major deformation mechanisms in pure Al and Ni are slip by Shockley partials, with possible twinning in both tensile and shear loading conditions. The Al/Ni bimetallic system underwent heat treatment at 1000 K, resulting in the rapid diffusion of Ni atoms into the Al matrix. When the holding time exceeded 10 ns, a stable Al_3_Ni intermetallic compound formed in the interfacial region.

In addition to examining the isolated effects of temperature or time on the formation of the Al/Ni interface, some researchers have explored the influence of electric fields, mechanical stresses, and multi-field coupling on the atomic diffusion behavior of the Al/Ni bimetallic system. Based on a MD model, Luo C et al. [80] investigated the effects of electric fields, thermal fields, mechanical stresses, and multi-field interactions on the atomic diffusion behavior in nanometer-thick multilayer films. The simulation results indicate that the thermal field induces interlayer diffusion by shortening the diffusion path while promoting intralayer diffusion; mechanical stress enhances interlayer atomic diffusion in the thickness plane by deforming atoms within the layer; the electric field inhibits intralayer diffusion by affecting the directional migration of metal atoms; and the combined effects of electric, thermal, and mechanical fields can suppress diffusion in the Al/Ni system. Moreover, the impact velocity also significantly affects the diffusion behavior at the Al/Ni interface.

### 4.2. Mechanical Behavior of Al/Ni Bimetallic Interfaces

#### Tensile Behavior

Internal Influencing Factors

Internal factors, such as crystallographic orientation, play a crucial role in determining the macroscopic mechanical properties of Al/Ni bimetallic materials. For instance, Qiao C et al. [81] investigated the evolution and mechanical behavior of Al/Ni bimetallic interfaces with [100], [111], and [110] orientations subjected to counterclockwise and clockwise torsion. The findings indicate that dislocation propagation velocity at the bimetallic interface is significantly reduced, with dislocations predominantly confined to the interface region. Furthermore, the shear stress at the bimetallic interface is notably greater than that at monometallic grain boundaries, as illustrated in Figure 25. Zheng DL et al. [82] investigated the influence of misfit dislocations on glide dislocations at the Ni/Al interface using MD simulations. The study revealed that different crystal orientations result in significantly varying interfacial mismatch dislocation structures, including square networks, disordered arrangements, and zigzag formations. Under uniaxial tensile loading, slip dislocations nucleate from the mismatch dislocation line and propagate primarily in the softer Al, predominantly moving along the {111} slip surface. Additionally, during expansion, slip dislocations may collide, leading to the release of dislocation energy. The results indicate that the Al(001)/Ni(111) system exhibits the highest critical stress at 2.4 GPa, while the Al(001)/Ni(110) system demonstrates the lowest critical stress at 1.9 GPa, attributed to the disordered interface and the random nucleation of slip dislocations.

External Influencing Factors

Hu ZJ et al. [72] also examined the impact of temperature on the mechanical properties of diffusion-bonded Al/Ni bimetals. The study revealed that the diffusion-bonded Al/Ni bimetal exhibits superior stiffness compared to single-crystal aluminum; however, its tensile yield strength is lower than that of single-crystal nickel or aluminum at equivalent temperatures. Moreover, the diffusion-bonded Al/Ni bimetal demonstrates a reduced Young’s modulus and tensile yield strength at elevated temperatures. During tensile loading, the material’s plastic deformation is primarily governed by dislocation activity at low temperatures and creep behavior at high temperatures, with the results depicted in Figure 26.

At present, no investigation on the compression behavior of the Al/Ni bimetal using MD simulations has been reported.

## 5. Molecular Dynamics Simulation of Al/Ti Bimetallic Interfacial Behavior 

### 5.1. Interface Formation Process of Al/Ti Bimetallic Systems

Internal Influencing Factors

Moon S et al. [83] studied the impact of sinusoidal interfaces of varying heights (h) and spatial periods (p), as well as that of flat interfaces, on the atomic diffusion of Al/Ti bimetals. The simulation results indicate that the shape of the interface affects the mechanical mixing of Al and Ti atoms: there is less mixing at the flat interface, while the sinusoidal interface promotes more active mixing. The degree of mechanical mixing of atoms at the sinusoidal interface increases with the height and decreases with the spatial period. Atom mixing primarily occurs in the direction of the vibrational loading (the xz-plane), while it is suppressed in the normal direction of the interface (the y-direction), as shown in Figure 27.

External Influencing Factors

Researchers focused on the Al/Ti system and used MD simulations to investigate atomic diffusion behavior from different process perspectives. Liu JY et al. [84] systematically analyzed the diffusion behavior of Ti and Al atoms across a range of temperatures, finding that the Ti atoms mainly diffused to the Al atoms as the diffusion temperature increased. A new phase was generated at the interface. This process was concurrently associated with increased lattice distortion at the interface and a significant growth in the thickness of the interdiffusion layer. Moreover, under fixed pressure and heating rate conditions, the diffusion coefficient demonstrated a temperature-dependent increase that aligns with the Arrhenius relationship. When the pressure was 30 MPa, the diffusion activation energy values of the Ti atoms and Al atoms in the ideal Ti/Al interface were 5.39 kJ/mol and 37.90 kJ/mol. They also pointed out that the diffusion effect of the rough interface was more evident, but the values of the diffusion coefficient and the activation energy were not very different from those of the ideal interfaces, as shown in Table 4.

Aside from the impact of single parameters, the diffusion process at the Al/Ti bimetallic interface is comprehensively controlled by multiple factors, including temperature and pressure. Tang FL et al. [85] analyzed the diffusion behavior of surface atoms during the diffusion welding of Al/Ti bimetals. The results showed that at 700 K and 50 MPa, the diffusion process mainly involved Ti atoms diffusing into Al, while the aluminum atoms rarely diffused into titanium. Wang HN et al. [86] studied the evolutionary process of the interface of Al/Ti bimetals during the hot pressing process. The results indicated that at 600 K and 50 MPa, the atomic amplitude of aluminum atoms was greater than that of titanium atoms during the bonding process, leading to higher instability. Aluminum and titanium atoms bonded near the interface to form a transition layer, and its thickness gradually increased over time, although the rate of increase decreased, exhibiting a nonlinear relationship. Additionally, from the analysis of mean square displacement, the diffusion coefficients of Al and Ti atoms were determined to be 3.55 × 10^−10^ and 2.73 × 10^−10^ m^2^·s^−1^, respectively.

Atomic diffusion not only modifies the interfacial microstructure but, more importantly, when reaching a critical concentration threshold, induces strong chemical interactions between Ti and Al atoms that lead to intermetallic compound formation. Malekpour F et al. [87] investigated the TiAl formation mechanism using MD simulations combined with spark plasma sintering (SPS). Their analysis revealed a sequential phase evolution: TiAl_3_ initially forms at the Al/Ti interface during atomic interdiffusion at 1340 K, followed by Ti_3_Al formation at the Ti/TiAl_3_ interface. Subsequent aluminum diffusion results in the formation of Ti_2_Al_5_ and TiAl_2_ at the Ti_3_Al/TiAl_3_ and Ti_3_Al/Ti_2_Al_5_ interfaces, respectively, with TiAl ultimately forming at the TiAl_2_/Ti_3_Al interface. Kiselev SP et al. [88,89] simulated the formation of TiAl_3_ and Ti_3_Al compounds during cold spraying of aluminum coatings onto titanium alloy substrates, followed by annealing in argon atmosphere at temperatures ranging from 830 K to 1270 K.

### 5.2. Mechanical Behavior of Al/Ti Bimetallic Interfaces

#### 5.2.1. Tensile Behavior

Internal Influencing Factors

AN MR et al. [90] systematically studied the effects of interface structure and interlayer spacing on the plastic deformation mechanisms of Al/Ti bimetal composites. Their results demonstrated that for coherent interfaces, increasing interlayer spacing induces two distinct transitions in the Ti layer deformation mechanism: from dislocation-dominated plasticity to HCP → FCC phase transformation, and subsequently, back to dislocation slip dominance. In contrast, non-coherent interfaces exhibited HCP → FCC phase transformation as the primary deformation mechanism regardless of interlayer spacing. In both interface types, aluminum layers predominantly deformed via confined layer slip (CLS) of dislocations. Notably, when interlayer spacing exceeded the optimal value of 6.6 nm, non-coherent interface samples displayed significant secondary hardening while maintaining superior toughness compared to coherent interfaces, as illustrated in Figure 28.

Lu XC et al. [91] focused on the impact of TiAl_3_ interlayer thickness on the mechanical properties of Al/Ti composites. Their MD simulations confirmed that while moderate TiAl_3_ thickness enhances plasticity through coordinated deformation of Ti/Al/TiAl_3_ layers, excessive thickness promotes localized stress concentration and high-strain band formation, thereby facilitating crack nucleation and reducing overall ductility.

External Influencing Factors

Wang HN et al. [86] investigated the effects of cooling temperature on the tensile behavior of Al/Ti interfaces. Their results revealed a distinct transition in the deformation mechanism: below 300 K, the material exhibited purely brittle fracture characteristics, while above 300 K, plastic deformation became increasingly prominent. The ultimate tensile strength (UTS) showed an inverse relationship with cooling temperature, and fracture consistently occurred in the aluminum bulk rather than at the well-bonded Al/Ti interface, as shown in Figure 29.

Lu XC et al. [91] also investigated the tensile deformation behavior of Al/Ti bimetals at elevated temperatures. MD simulations confirmed a reduction in the elastic modulus and peak stress at high temperatures. The elastic modulus and peak stress of the Al/Ti bimetal exhibited a decreasing trend with increasing temperature, which is consistent with tensile experimental results. However, the influence of temperature on flow stress was found to be minimal. Additionally, the coordinated deformation of the Ti, Al, and TiAl_3_ layers at high temperatures contributes to the enhanced plasticity of the Al/Ti laminate.

Moreover, beyond the individual effects of internal or external factors on the deformation mechanism, the combined influence of internal and external factors can significantly alter the material’s deformation behavior. For instance, An MR et al. [92] examined the effects of interlayer spacing and temperature on the deformation mechanism of Ti (0001)/Al (111) bimetals. The results indicated that at 0.01 K and 300 K, stress concentration-driven interface rotation dominated the plastic deformation of samples with smaller layer thicknesses. At 0.01 K, larger thickness samples exhibited constrained layer slip of dislocations in the Al layer, as well as the formation of basal/prism interfaces and a phase transition from HCP-Ti to FCC-Ti in the Ti layer. Additionally, it was observed that at 300 K, restricted layer slip and necking of dislocations in the Al layer were potential deformation mechanisms for samples with greater thicknesses.

#### 5.2.2. Compression Behavior

The mechanical behavior and deformation mechanisms during the compression process of materials are not only closely related to the atomic arrangement and crystal structure within the material, but are also significantly influenced by various external factors.

Internal Influencing Factors

Similar to tensile loading, different processing parameters also have a significant impact on the compressive performance of Ti/Al bimetals. Chen SQ et al. [93], utilizing MD simulations, studied the effects of temperature and Ti volume fraction on the compression deformation of Al/Ti bimetals. The findings indicated that during the compression process, analogous to the tensile deformation, both the elastic modulus and peak stress of the material decreased with increasing temperature; furthermore, the compressive strength decreased as temperature rose, leading to increased plasticity of the microstructure. Additionally, the compressive strength increased with the rising Ti volume fraction. Upon reaching the compressive strength, the initial plastic deformation and dislocations in samples A and B were primarily concentrated on the Ti side, with a predominant dislocation type of 1/6<112>; meanwhile, the plastic deformation of sample C was mainly concentrated on the Al side. Notably, sample B exhibited the best compressive performance, as illustrated in Figure 30.

## 6. Molecular Dynamics Simulation of Al/Fe Bimetallic Interfacial Behavior 

### 6.1. Interface Formation Process of Al/Fe Bimetallic Systems

A significant quantity of brittle and hard intermetallic compounds forms at the Al/Fe bimetallic interface, which substantially degrades the mechanical properties of the joint [120]. Achieving a comprehensive understanding of the interfacial atomic diffusion behavior and the microscopic formation mechanism of intermetallic compounds holds guiding significance for enhancing the preparation process and optimizing material properties [121].

External Influencing Factors

Chlouk ZE et al. [94] investigated the interfacial diffusion of the Al/Fe system under uniaxial compression at various temperatures using MD simulations. The simulation results demonstrate that dislocation motion is accompanied by significant interfacial mixing under compression and temperature conditions. Further analysis reveals that higher stress levels exert a more pronounced effect on atomic diffusion. Even at relatively low temperatures, Al and Fe atoms exhibit notable interfacial mixing under applied high-pressure stress, as depicted in Figure 31. This solid-phase mixing leads to the formation of FeAl intermetallic compounds (CsCl crystal structures). In contrast, Luo Q et al. [95] identified two intermetallic compounds, Fe_4_Al_13_ and Fe_2_Al_5_, at the Al/Fe interface, with Fe_4_Al_13_ being the dominant phase. Thermodynamic calculations and MD simulations were employed to validate its potential mechanism.

The incorporation of heterogeneous metals effectively suppresses the growth of intermetallic compounds, making it particularly suitable for the efficient metallurgical bonding of bimetals. The issue of insufficient interface bonding capability due to the disparity in thermal and physical properties between Al and Fe is especially prominent. Consequently, Liu SY et al. [96] adopted electrodeposition of a Cu layer on the iron substrate as the Al/Fe bimetallic intermediate layer combined with hot plating and composite casting to address these challenges. The influence of Cu interfacial thickness on the bimetallic interfacial structure and shear strength was examined. The experimental results reveal that the shear strength of Al/Fe bimetallic materials initially increases and subsequently decreases with increasing Cu interlayer thickness, reaching a maximum of 77.65 MPa. The introduction of the Cu interlayer enhances interface bonding performance.

### 6.2. Mechanical Behavior of Al/Fe Bimetallic Interfaces

#### Compression Behavior

External Influencing Factors

El Chlouk ZG et al. [97] explored the effects of strain rate and temperature on the mechanical behavior of the Al/Fe interface under compressive loading through MD. The results indicate that at temperatures below 500 K, the stress–strain curves exhibit two distinct yield points corresponding to dislocation nucleation in Al and Fe, respectively. As the temperature rises, these two yield points gradually merge into one. Both yield stress and flow stress decrease with increasing temperature and increase with rising strain rate. The interaction of these two factors adheres to the thermal activation model, as illustrated in Figure 32.

The study of the tensile properties of the Al/Fe bimetallic interface has not been adequately explored, highlighting the urgent need for systematic research to fill this gap.

## 7. Molecular Dynamics Simulation of Cu/Ni Bimetallic Interfacial Behavior 

### 7.1. Interface Formation Process of Cu/Ni Bimetallic Systems

The research on the formation process of the Cu/Ni bimetallic interface mainly focuses on external factors, while there are relatively few studies on internal factors.

External Influencing Factors

Extensive research has been conducted on the diffusion behavior at Cu/Ni bimetallic interfaces, primarily focusing on shear–annealing coupling effects and temporal influences. Chen NJ et al. [98] employed MD simulations to investigate the mechanisms of defect substructure evolution and atomic diffusion in a nanocrystalline Cu/Ni bimetal under severe shear–annealing coupling. The simulations revealed that under intense shear, triple junction sliding facilitates long-range atomic mixing through net dislocation flux, while interfacial dislocation sliding induces short-range mixing. Subsequent annealing treatments further promote atomic mixing through dynamic evolution of non-equilibrium defect substructures, as detailed in Figure 33.

Sun JX et al. [99] investigated the high-temperature diffusion process of nano-Cu/Ni bimetallic structures using MD simulations. They performed tensile simulations on the diffusion models obtained after annealing and analyzed the effects of the holding time on the interface structure and mechanical properties. The results indicate that at 1600 K, the longer the holding time, the greater the thickness of the transition layer, and the lower the tensile strength of the diffusion model. A fracture occurred within the diffusion layer in the diffusion model. When the holding time was 600 ps, the tensile strength of the diffusion model reached 11.62 GPa, which corresponded to 76% of the tensile strength of the ideal contact Cu/Ni model.

Pang WW et al. [100] examined the influence of Fe solute atoms on the interfacial structure of Cu/Fe_x_Ni_1−x_ layered composites. Their results demonstrate that when the Fe content is below 70%, the Fe_x_Ni_1−x_ layer maintains an FCC structure with triangular FCC/FCC interfaces. At Fe concentrations exceeding 70%, the Fe_x_Ni_1−x_ layer transforms into a BCC structure, forming elongated FCC/BCC interfaces, as shown Figure 34. Increasing the Fe content induces an interfacial transition from semi-coherent to incoherent configurations, with stress concentration zones shifting from the Cu layer to the Fe_x_Ni_1−x_ layer and eventually returning to the Cu layer.

### 7.2. Mechanical Behavior of Cu/Ni Bimetallic Interfaces

#### Tensile Behavior

Current research on Cu/Ni bimetal interfacial mechanics predominantly focuses on internal factors, such as interface structure, tilt angles, twist angles, and defects. Comparatively fewer studies address external factors, mainly limited to strain rates and alloying element additions.

Internal Influencing Factors

Numerous researchers have investigated how the interfacial structure affects the mechanical properties of Cu/Ni bimetals. Gang C et al. [101] performed MD simulations of uniaxial tension in a Cu/Ni bimetal with coherent and semi-coherent interfaces. Their results show that during deformation, dislocations nucleate at interfaces, with leading dislocations crossing interfaces to initiate plastic deformation. Semi-coherent interfaces impede dislocation glide, making the strength and strain rate sensitivity of nanoscale Cu/Ni bilayers thickness-dependent. In contrast, coherent interfaces offer weak barriers, allowing easy dislocation transmission and rendering mechanical properties thickness-independent.

Twin boundary structures also significantly influence Cu/Ni mechanical performance. Weng SY [102] conducted MD simulations at 10 K, 100 K, and 300 K for Cu/Ni with coherent, semi-coherent and coherent twin boundaries. The simulations revealed that coherent twin boundaries consistently provide remarkable strengthening regardless of temperature, while coherent and semi-coherent interfaces show negligible strengthening effects. Fu T et al. [103] compared pure Cu, pure Ni, and a Cu/Ni bimetal under cylindrical indentation through MD, particularly examining cube–cube interfaces and hetero-twin boundaries. Their findings indicate that hetero-twin boundaries facilitate bimetal hardening, with increasing boundary density enhancing Cu/Ni hardness.

Interface tilt angles represent another critical factor. Yang M et al. [104] studied a Cu/Ni bimetal with interfaces that were parallel, tilted, and perpendicular to the tensile direction. The 45° tilted sample exhibited optimal plasticity, with yield stress showing a V-shaped dependence on tilt angle, reaching minimum at 33°, as shown in Figure 35a. Additionally, yield strength increased with layer thickness due to dislocation interactions across different slip planes, as shown in Figure 35b.

Chen SD et al. [105] investigated Cu(001)/Ni(001) interfaces with various twist angles under uniaxial loading. Yield strength initially decreased with increasing twist angle, reaching a minimum at 5.906°, then increased to a maximum at 15.124°, before stabilizing beyond 20°.

Wu CD et al. [106] studied defect-free versus defective (voids/protrusions) Cu/Ni interfaces under tension and shear using embedded-atom potentials. The defect-free bimetal demonstrated superior ultimate strength and strain during tensile testing.

External Influencing Factors

Multiple studies have employed MD to examine Cu/Ni mechanical properties under varying strain rates and alloying conditions.

Cheng C et al. [107] conducted simulations to investigate the deformation behavior and mechanical properties of nano-polycrystalline Cu/Ni bimetallic materials under varying strain rates. These findings indicate that the yield strength of nanocrystalline Cu/Ni films significantly increases with increasing strain rate, demonstrating a higher sensitivity to strain rate changes. Under low strain rate loading conditions, voids were observed at the Cu/Ni interface. In contrast, under high-strain-rate impact loading, the Cu layer failed in the form of fragmentation.

Certain studies have confirmed that the incorporation of heterogeneous elements significantly enhances the interfacial bonding strength between dissimilar materials through the formation of chemical bonds and metallic transition layers, thereby substantially improving the mechanical properties of composite materials [122]. Pang WW et al. [100] systematically investigated the influence of Fe solute atoms on both the interfacial structure and deformation behavior of Cu/Fe_x_Ni_1−x_ layered composites. Their comprehensive analysis revealed a remarkable finding: the ultimate stress demonstrates a complex non-monotonic relationship with the Fe content. Specifically, the maximum ultimate stress of 17.4 GPa was achieved when the Fe concentration ranged between 20 and 30%, representing a significant 15.2% enhancement compared to conventional Cu/Ni bimetal systems. Furthermore, their study observed a progressive reduction in interface shear strain width with increasing Fe content, as illustrated in Figure 36. This inverse correlation suggests that Cu/Fe_x_Ni_1−x_ interfaces with lower Fe concentrations exhibit superior plastic compatibility and deformation coordination capabilities.

However, there have been no reported studies on the compression properties of Cu/Ni bimetals.

## 8. Molecular Dynamics Simulation of Fe/Cu Bimetallic Interfacial Behavior 

### 8.1. Interface Formation Process of Fe/Cu Bimetallic Systems

External Influencing Factors

During the formation of Fe/Cu bimetal composites, atomic diffusion at the interface is an inevitable consequence of the concentration gradient and chemical potential difference between Fe and Cu atoms. Numerous researchers [108,109,110] have systematically investigated the diffusion behavior in Fe/Cu bimetals. Simulation results consistently demonstrate that Cu atoms exhibit higher diffusion coefficients than Fe atoms under all temperature and time conditions, although Fe atoms predominantly diffuse into the Cu side during diffusion bonding.

However, significant discrepancies exist among research findings regarding key aspects. Zhang GW et al. [108,109] established a model predicting the temperature dependence of diffusion distance, identifying 1473–1753 K as the optimal temperature range for Fe/Cu interfacial diffusion bonding. Their work revealed that vacancy mechanisms dominate the diffusion process. Notably, the addition of Ni enhanced both diffusion coefficients and distances of Fe/Cu atoms compared to initial states, with the diffusion distance showing a non-monotonic relationship with the Ni content. In contrast, Zheng HY et al. [110] reported that diffusion coefficients and distances increase monotonically with temperature and time, reaching a maxima at 1523 K after 3 ns. The diffusion coefficient of Fe atoms is in the order of Fe (100) < Fe (110) < Fe (111), and the diffusion coefficient of Cu atoms is in the order of Cu (110) > Cu (111) > Cu (100).

Common fabrication methods for Fe/Cu bimetals include explosive welding, diffusion bonding, powder metallurgy, and casting composites [111,112]. Increasing the impact velocity resulted in a larger area of metallurgical bonding, thereby enhancing the bonding strength [123]. Feng JR et al. [113] employed MD to investigate the effects of collision velocity (1000–2000 m/s) on interface formation during explosive welding, modeling three stages: loading, unloading, and cooling. Their results demonstrate velocity-dependent mechanisms: solid-state bonding dominates at lower velocities (1000–1500 m/s) through high-pressure effects below melting points, while higher velocities (1750–2000 m/s) induce melting, forming diffusion layers and nanocrystals that enhance mechanical properties, as shown in Figure 37. Their study clarified that melting is not essential for bonding, with mechanisms transitioning from pressure welding (solid-state) to fusion–diffusion welding (melt-assisted).

### 8.2. Mechanical Behavior of Fe/Cu Bimetallic Interfaces

#### Tensile Behavior

Internal Influencing Factors

Through comprehensive MD simulations, Shen Y J et al. [114] established 10 well-defined interface models to examine Fe/Cu interfacial characteristics and dislocation evolution mechanisms. The Cu($\overset{-}{1}1\overset{-}{1}$)/Fe(110)-KS and Cu($\overset{-}{1}1\overset{-}{1}$)/Fe(110)-NW interfaces exhibited superior performance, combining low interfacial energy with high mechanical strength compared to other orientations.

External Influencing Factors

Shen YJ et al. [114] performed a systematic investigation of the anisotropic mechanical response in Fe/Cu bimetallic interfaces under different loading orientations. Their results reveal that interface-mediated dislocation blocking mechanisms significantly enhance the tensile strength of Cu when loaded parallel to the interface plane. In contrast, perpendicular loading configurations exhibit reduced tensile strength compared to monolithic Cu, attributable to weaker interfacial cohesion, with quantitative data presented in Table 5. These simulation results show excellent agreement with existing experimental observations.

Notably, strain rate demonstrates a significant influence on the mechanical properties of Fe/Cu bimetals. Zheng HY [110] conducted MD simulations to investigate the mechanical behavior of Fe/Cu solid–liquid interfaces under the coupled effects of strain rate and temperature. The results reveal that the interfacial yield strength and fracture strain exhibit positive strain-rate sensitivity but negative temperature dependence. Specifically, at a strain rate of 1 × 10^9^ s^−1^ and tensile temperature of 300 K, the Fe(110)/Cu(110) diffusion interface achieves a yield strength of 12.1 GPa and a fracture strain of 21%. A comparative analysis indicates that the diffusion-processed interface contains substantially fewer stacking faults and dislocations than its non-diffusion counterpart, resulting in reduced tensile yield strength. Furthermore, Lin ZJ et al.’s [115] simulations on strain rate effects in Cu/Fe composites demonstrated monotonic increases in both yield stress and flow stress with increasing strain rate.

Alloying element incorporation can substantially modify Fe/Cu interfacial properties through solid solution strengthening or intermetallic formation. Zhang GW et al. [108] characterized the non-monotonic Ni-content dependence of yield strength in Fe/Cu-Ni systems, elucidating the underlying strengthening/weakening mechanisms. Through MD simulations, Wang S et al. [116] investigated Al-mediated grain boundary segregation effects in Fe/Cu embrittlement systems. Their findings demonstrate that Al exhibits faster grain boundary diffusion kinetics than Cu, with preferential segregation forming effective diffusion barriers that suppress Cu penetration. While Cu doping severely degrades Fe grain boundary cohesion, Al doping produces the opposite effect, as quantitatively shown in Figure 38.

## 9. Conclusions and Perspectives

### 9.1. Conclusions

A systematic comparison and summary of diffusion coefficients across various bimetallic systems are presented in Table 6. Diffusion coefficients for the Al/Cu bimetal system under varying conditions are summarized in Table 1.

Table 7 summarizes the research directions and progress of bimetals in recent years. From Table 6, it can be found that, at present, the studies on the interfacial behavior of bimetals are mainly concentrated in the fields of Al/Cu, Al/Mg, Al/Ni, Al/Ti, Al/Fe, Cu/Ni, and Fe/Cu. These studies mainly investigate the interfacial atomic diffusion behavior and its related properties, including diffusion coefficients, diffusion depths, and diffusion mechanisms. In addition, the studies focus on the microscopic mechanisms of the thickness and formation of intermetallic compounds, as well as the structural evolution process. Meanwhile, plastic deformation mechanisms, tensile properties, fracture modes, and interfacial strengthening mechanisms have also been the focus of previous studies. These studies provide important theoretical foundations and experimental data for understanding and optimizing the properties of bimetallic materials.

### 9.2. Perspectives

The molecular dynamics simulations conducted in this study offer comprehensive insights into the fundamental properties of various bimetallic systems, with a particular focus on their interfacial bonding strength and atomic diffusion behavior. These findings are pivotal for advancing the performance optimization of related materials for practical applications. Specifically, in the context of the Al/Mg system, understanding its exceptional lightweight–strength combination and interfacial behavior, such as the manner in which the high-entropy alloy coating effectively inhibits diffusion between Al and Mg, thereby preventing the formation of Al/Mg intermetallic compounds, will directly inform the development of lighter and more durable transportation components, such as automotive hubs and body panels, ultimately enhancing fuel efficiency and safety. Consequently, this study not only enriches the scientific understanding of bimetallic interfaces but also establishes a critical theoretical foundation for improving the performance and reliability of bimetallic materials in everyday applications, such as lightweight transportation.

In the MD simulation studies of bimetallic interfacial behavior, although some progress has been made, there are still many research directions that need to be further explored and improved. With the improvement of computational power and the continuous development of simulation technology, future research can be deepened in several aspects. The possible future research directions are as follows:(1)Machine learning interatomic potentials to optimize potential functions: The key to materials simulation lies in the interatomic potentials, which are crucial for accurately predicting and understanding the physicochemical properties of materials. Classical MD simulations usually rely on experimental data or potential energy surfaces fitted by approximate methods to describe the atomic forces, and although they can match experimental results in many cases, their applicability and accuracy are still limited, especially in complex solid-liquid systems where it is difficult to describe the bonding and breaking of chemical reactions. Therefore, the construction and optimization of interatomic potentials using machine learning methods has become an effective means to describe the atomic-scale behavior of materials more accurately.(2)Applied energy field-assisted interfacial bonding: It mainly includes explosion bonding and electromagnetic pulse bonding. Explosions or electromagnetic pulses produce transient tilted collisions and plastic deformations at the composite interface, thus enhancing the interfacial bonding. MD simulations can provide an in-depth analysis of the effects of these applied energy fields on the interfacial atomic diffusion, interfacial structure evolution, and bond strength, thus providing new ideas and methods for optimizing the properties of bimetallic materials.(3)Improvement of stress–strain curves: When using LAMMPS version 29 August 2024 Stable Release to carry out simulation studies, it was found that there was a certain degree of difference between the obtained stress–strain curves and the corresponding curves obtained through actual experiments, which was manifested in the fact that the curves obtained from the simulation showed a wavy pattern at the final stage. In view of this situation, suitable fitting algorithms can be used to re-fit the stress–strain curve based on machine learning and other means, so that it can better reproduce the mechanical behavior of the experiment, thus improving the reliability and scientific validity of the simulation.(4)Multi-scale simulation methods: The interfacial behavior of bimetallic systems represents a complex multiscale and multiphysics process, necessitating cross-scale simulation analysis. To overcome inherent scale limitations in simulations, an integrated computational framework combining DFT calculations, MD simulations, phase-field methods, and finite element modeling has been established. This approach enables the following: a DFT calculation of Al/Mg interfacial binding energy; MD simulations of dynamic interface evolution to provide diffusion barriers and interfacial energy parameters for mesoscale models; phase-field simulations of matrix/interface microstructural evolution yielding equivalent mechanical properties for finite element inputs; and ultimately, finite element modeling based on mesoscale features to predict mechanical responses of Al/Mg bimetal—thereby achieving coordinated cross-scale design of interfacial performance.

## Figures and Tables

**Figure 1 materials-18-03048-f001:**
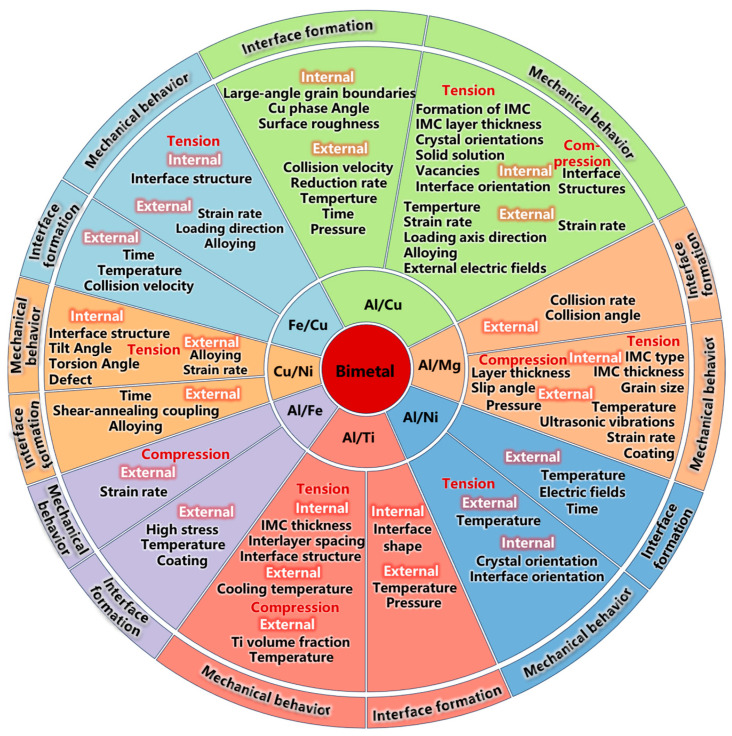
Behavioral characteristics of bimetallic interfaces under the influence of internal and external factors [39,40,41,42,43,44,45,46,47,48,49,50,51,52,53,54,55,56,57,58,59,60,61,62,63,64,65,66,67,68,69,70,71,72,73,74,75,76,77,78,79,80,81,82,83,84,85,86,87,88,89,90,91,92,93,94,95,96,97,98,99,100,101,102,103,104,105,106,107,108,109,110,111,112,113,114,115,116].

**Figure 2 materials-18-03048-f002:**
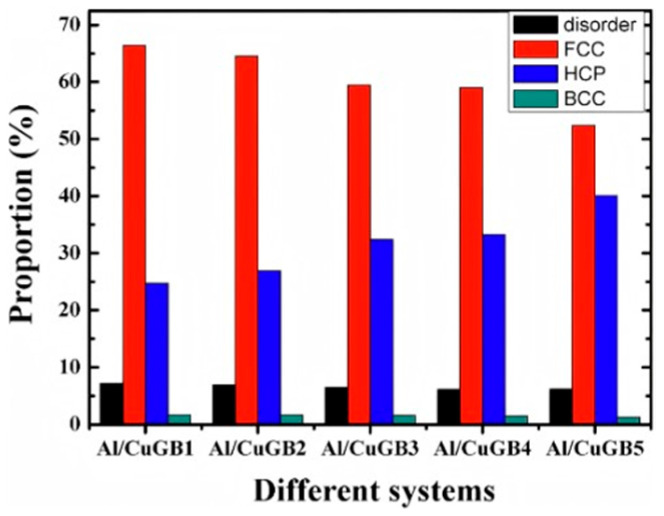
The structural proportion of the different CuGB/Al systems frozen at cooling rates of 0.1 K/ps [40].

**Figure 3 materials-18-03048-f003:**
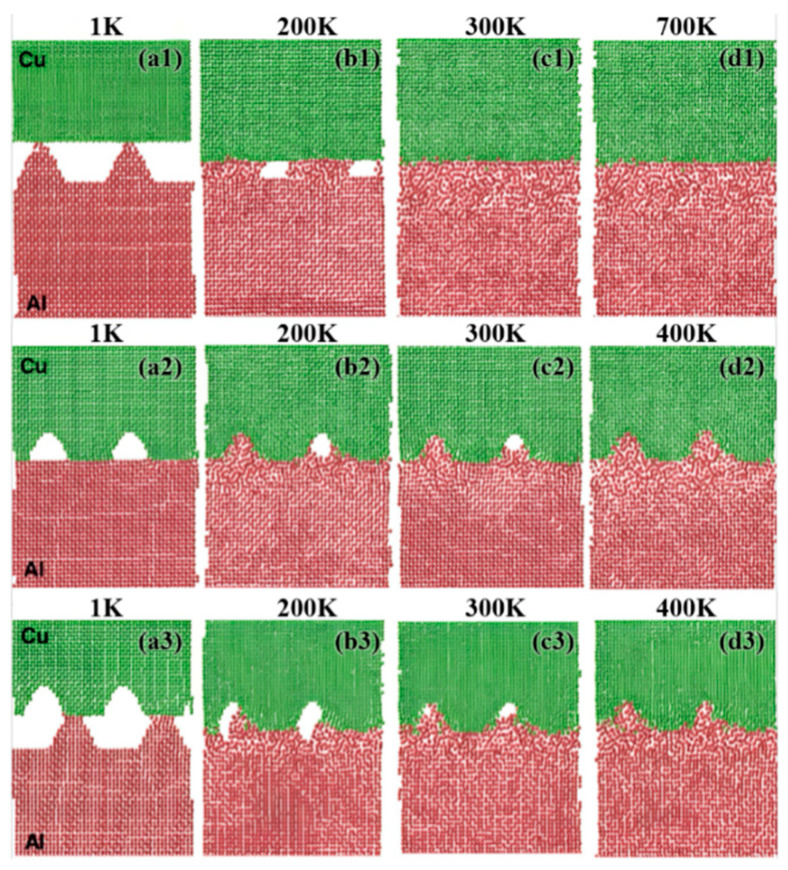
Deformed configurations of a cross-section at different temperatures during heating: (**a1**–**d1**) smooth Cu surface and rough Al surface, (**a2**–**d2**) smooth Al surface and rough Cu surface, (**a3**–**d3**) both surfaces are rough [41].

**Figure 4 materials-18-03048-f004:**
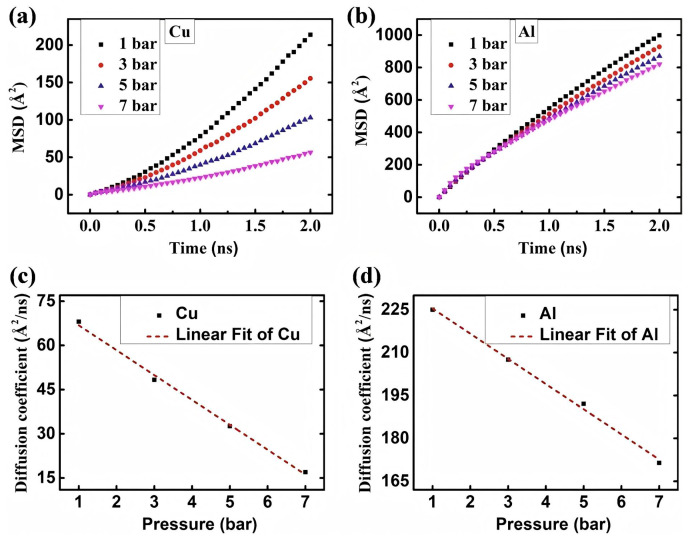
The MSD of vertical directions at different temperatures of (**a**) Cu and (**b**) Al and the Arrhenius plot for (**c**) Cu and (**d**) Al under 3 bar [44].

**Figure 5 materials-18-03048-f005:**
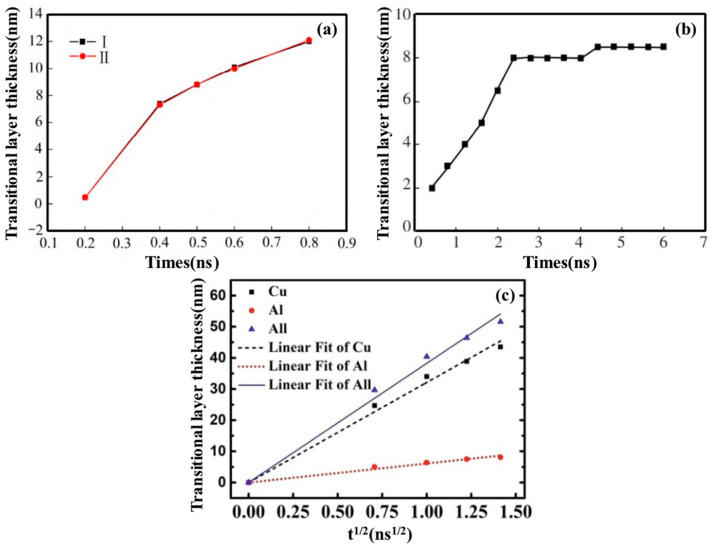
(**a**) Variation curve of thickness of transition layer with casting time [45], (**b**) change curve of thickness of the transition layer with time [42], (**c**) parabolic relationship between diffusion depth and diffusion time of t^1/2^ [44].

**Figure 6 materials-18-03048-f006:**
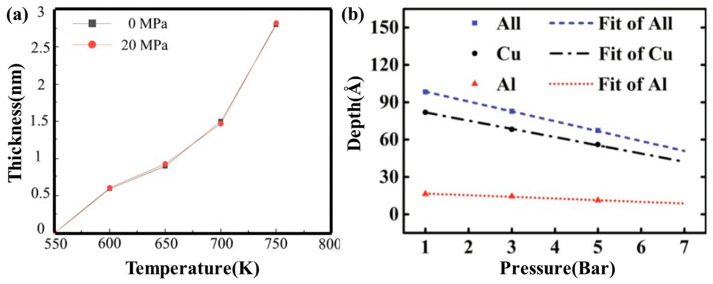
(**a**) Thicknesses of interfacial region at different temperatures and pressures [47], (**b**) diffusion layer thickness and pressure relationship diagram [44].

**Figure 7 materials-18-03048-f007:**
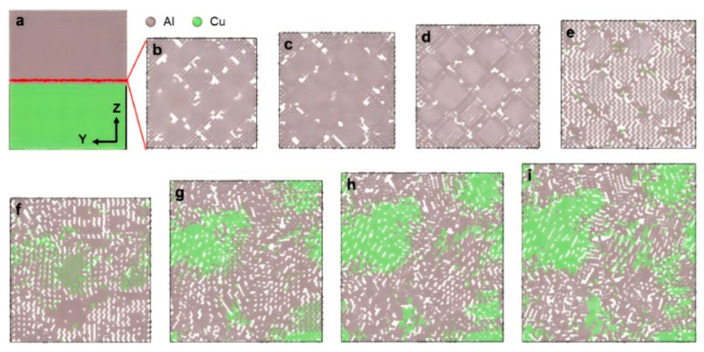
Schematic of the rolling process profile. (**a**) *Z*-axis is the position of the cross-section at 60–62 Å. Cross-section changes at a reduction rate of (**b**) 0, (**c**) 0.1, (**d**) 0.2, (**e**) 0.3, (**f**) 0.4, (**g**) 0.5, (**h**) 0.55, and (**i**) 0.6 [49].

**Figure 8 materials-18-03048-f008:**
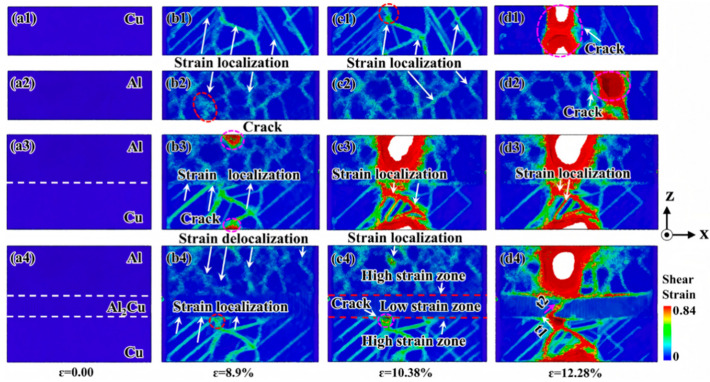
The strain distributions in the tensile direction: (**a1**–**d1**): nano-polycrystalline Cu, (**a2**–**d2**): nano-polycrystalline Al, (**a3**–**d3**): Cu/Al layered composites, (**a4**–**d4**): Cu/Al_2_Cu/Al layered composites [50].

**Figure 9 materials-18-03048-f009:**
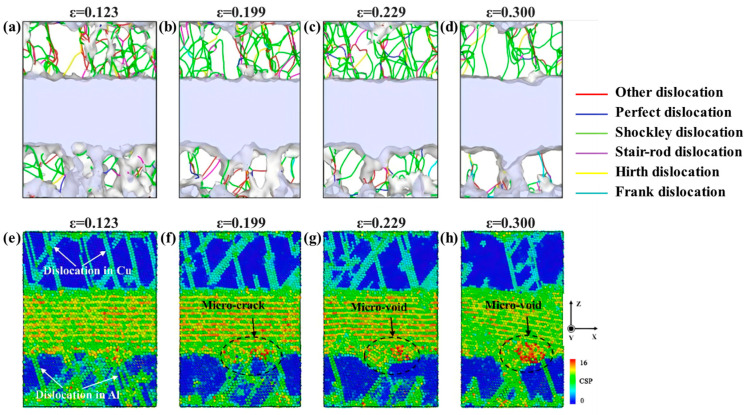
The Cu(111)/Al_2_Cu($\overset{-}{1}10$)/Al(111) composite corresponding to the yield stage II mark in the tensile stress–strain curve: (**a**–**d**) DXA, (**e**–**h**) centrosymmetry parameter method (CSP) [52].

**Figure 10 materials-18-03048-f010:**
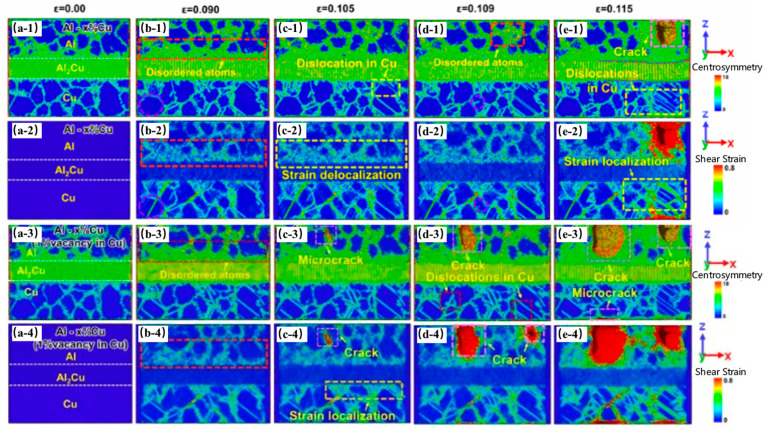
The Cu(Al)/Al_2_Cu/(Cu)Al layered gradient heterostructure with gradient solid solubility: (**a-1**–**e-1**): microstructural evolution, (**a-2**–**e-2**): local strain transfer in the Cu(Al)/Al_2_Cu/(Cu)Al layered gradient heterostructure with vacancies, (**a-3**–**e-3**): the microstructural evolution, (**a-4**–**e-4**): local strain transfer [53].

**Figure 11 materials-18-03048-f011:**
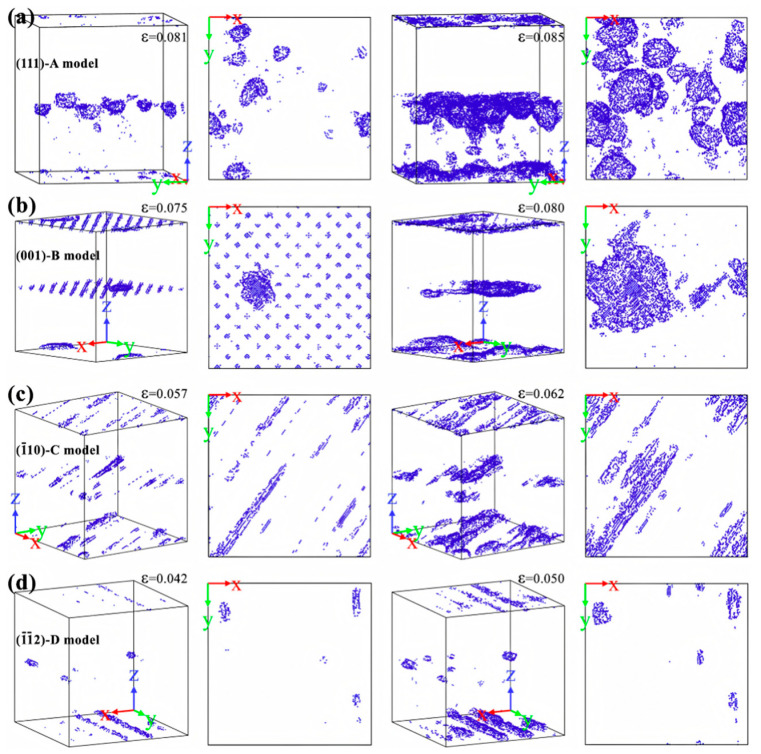
Nucleated voids inside the simulation box for four models with interface orientations: (**a**) [111], (**b**) [001], (**c**) [$\overset{-}{1}10$], and (**d**) [$\overset{-}{1}\overset{-}{1}2$] [54].

**Figure 12 materials-18-03048-f012:**
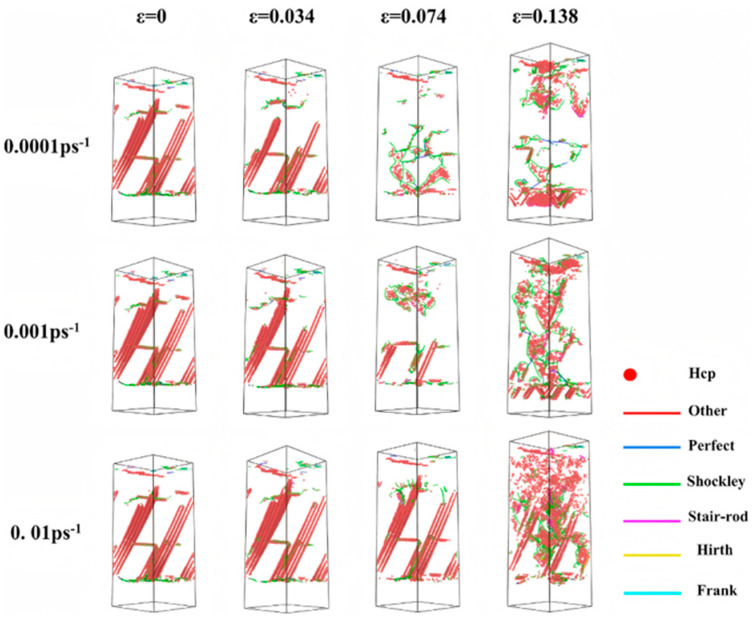
Structure and stress evolution diagram of Cu–Zr/Al interface model subjected to tensile load at different strain rates [56].

**Figure 13 materials-18-03048-f013:**
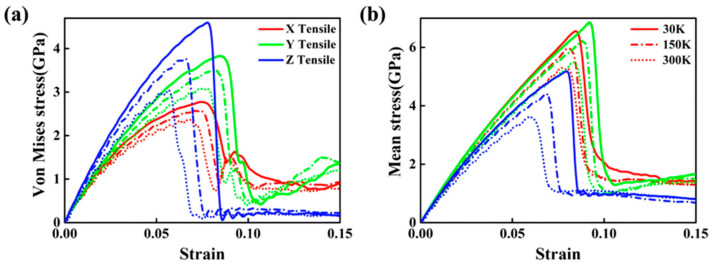
Stress–strain curves under different loading axes and temperatures: (**a**) von Mises stress, (**b**) mean stress [58].

**Figure 14 materials-18-03048-f014:**
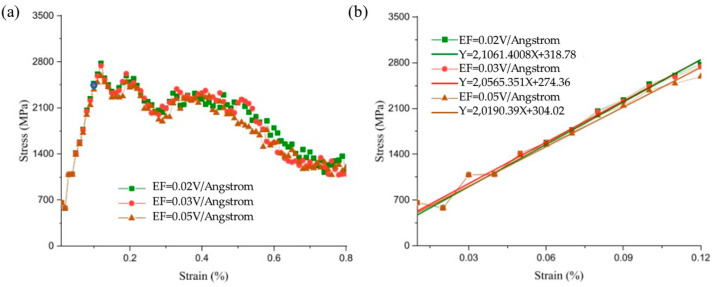
(**a**) The stress–strain curve and (**b**) YM of the simulated Al-Cu-Al TLNC at different EF values [59].

**Figure 15 materials-18-03048-f015:**
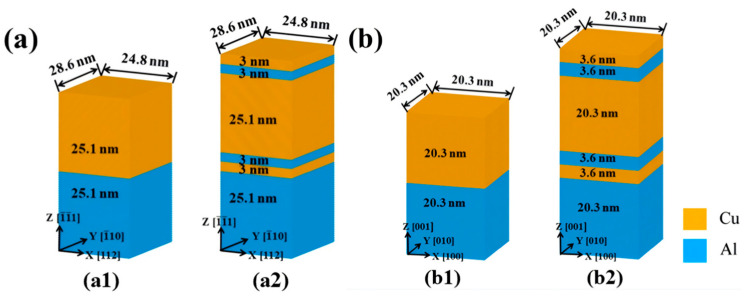
(**a**) The computational models of Al/Cu nanoscale metallic multilayers with a ($\overset{-}{1}\overset{-}{1}1$) interface: (**a1**) homogeneous sample, (**a2**) heterogeneous sample; (**b**) the computational models of Al/Cu nanoscale metallic multilayers with a (001) interface: (**b1**) homogeneous sample, (**b2**) heterogeneous sample [60].

**Figure 16 materials-18-03048-f016:**
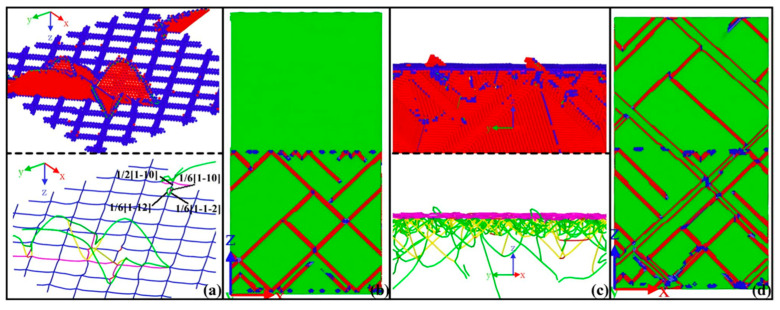
The evolution snapshots of atomic structures during the compression process for the loading strain rate of 10^8^ s^−1^. (**a**,**b**) Perfect misfit dislocations on the interface split into stair–rod dislocations and Shockley partial dislocations which quickly propagate inside the Al layer. (**c**) A partial dislocation crosses the interface into the Cu layer or nucleates at the interface and propagates into the Cu layer. (**d**) Dislocation slip and twinning dominate the plastic deformation during the process [61].

**Figure 17 materials-18-03048-f017:**
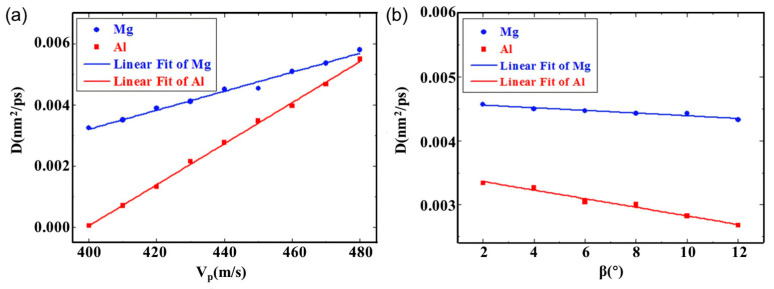
Diffusion coefficients of Al and Mg: (**a**) at different collision velocities for β = 10° and (**b**) at different collision angles for V_p_ = 440 m/s [62].

**Figure 18 materials-18-03048-f018:**
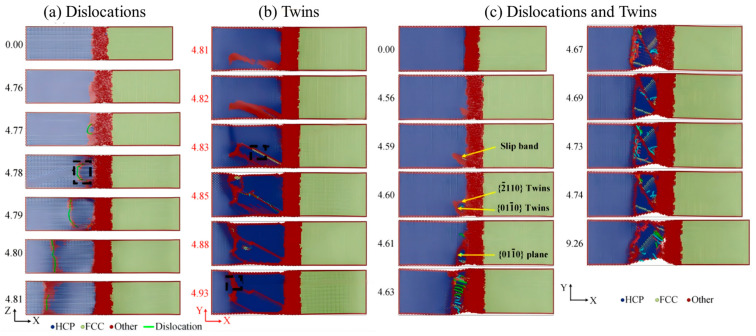
Yielding deformation of Mg-Al CS1 and CS2 at different strains (%): (**a**) motions of dislocations in CS1, (**b**) formations of twins in CS1, and (**c**) deformations of twins and motions of dislocations in CS2 [63].

**Figure 19 materials-18-03048-f019:**
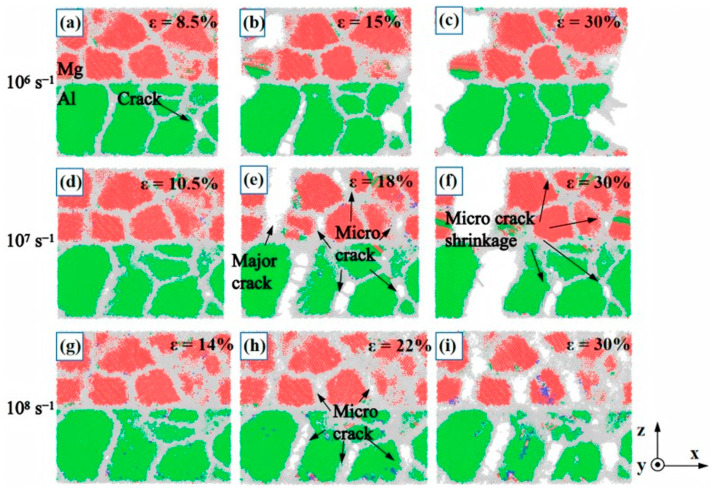
Crack initiation and propagation for nano-polycrystalline Al/Mg bimetals at different strain rates of (**a**–**c**) 10^6^ s^−1^, (**d**–**f**) 10^7^ s^−1^, and (**g**–**i**) 10^8^ s^−1^ [64].

**Figure 20 materials-18-03048-f020:**
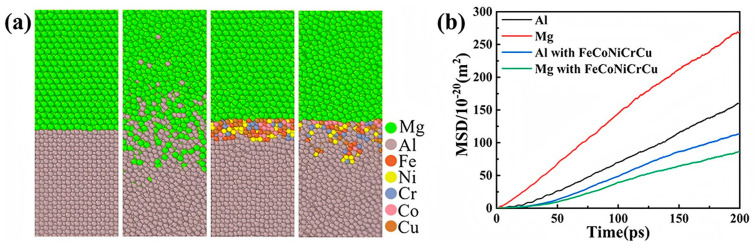
Molecular dynamics simulation results for two models: (**a**) atomic diffusion behavior at 0 ps and 200 ps and (**b**) MSD curves for Al and Mg atoms under different models [67].

**Figure 21 materials-18-03048-f021:**
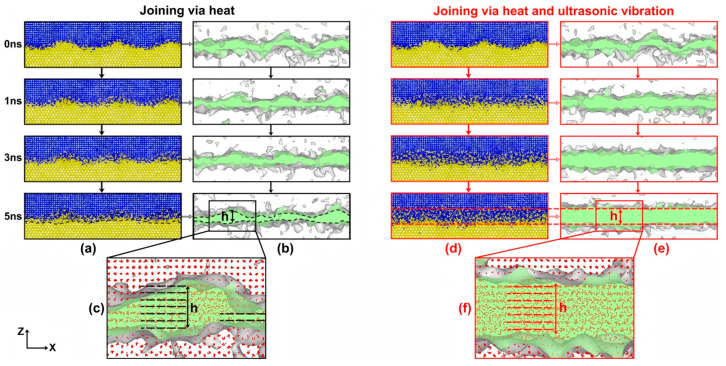
The joining of the Mg-Al nanolayer at Te = 800 K via (**a**) only heat and (**d**) heat and Uvs, where B = 5 nm and f = 5.7 GHz. (**b**,**c**,**e**,**f**) are the corresponding defect meshes of the Mg/Al interfaces. h is the thickness of the interface [68].

**Figure 22 materials-18-03048-f022:**
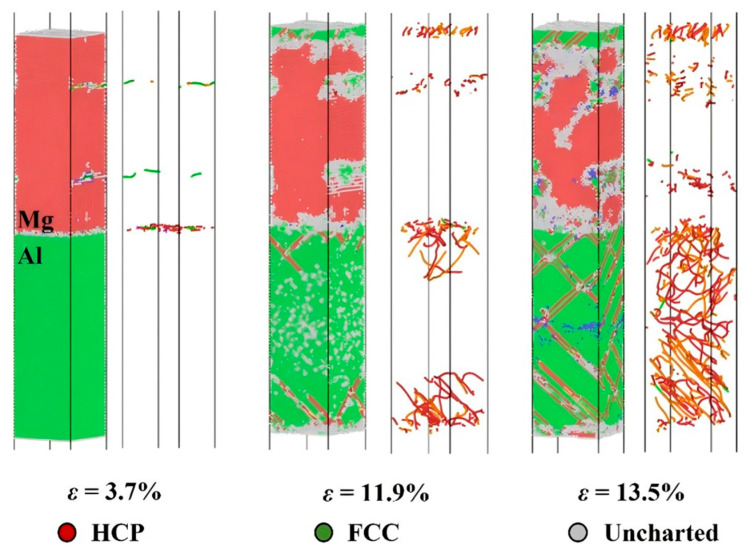
Dislocation extension and deformation of Al/Mg composite with a layer thickness of 26.7 nm at θ = 0° [69].

**Figure 23 materials-18-03048-f023:**
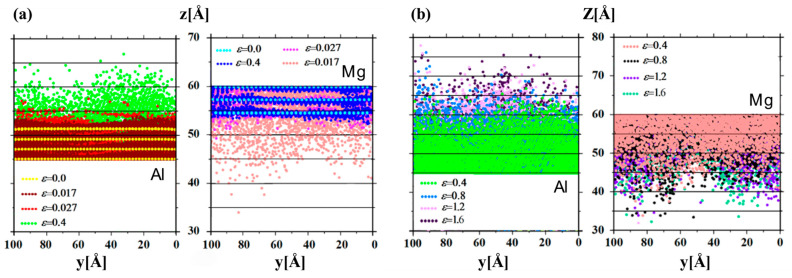
Changes of the atomic positions through Al−Mg interface: (**a**) for 0.0 < ε_xy_ < 0.4 and (**b**) for 0.4 < ε_xy_ < 1.6. Different colors correspond to different deformation stages. Only part of the sample is presented [70].

**Figure 24 materials-18-03048-f024:**
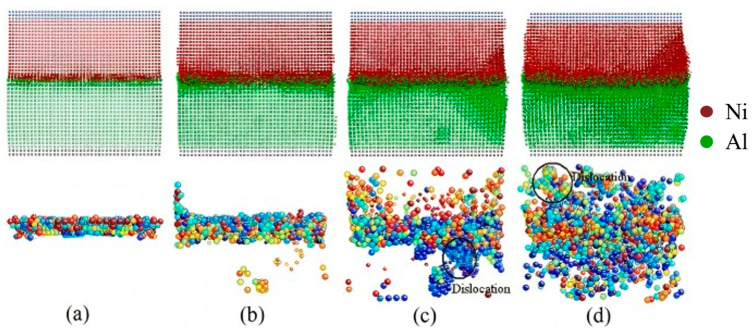
Planar views of diffusion-bonded Ni/Al nanowires (**top row**) and Ni/Al interfaces (**bottom row**): (**a**) 1 K, (**b**) 300 K, (**c**) 500 K, and (**d**) 700 K [72].

**Figure 25 materials-18-03048-f025:**
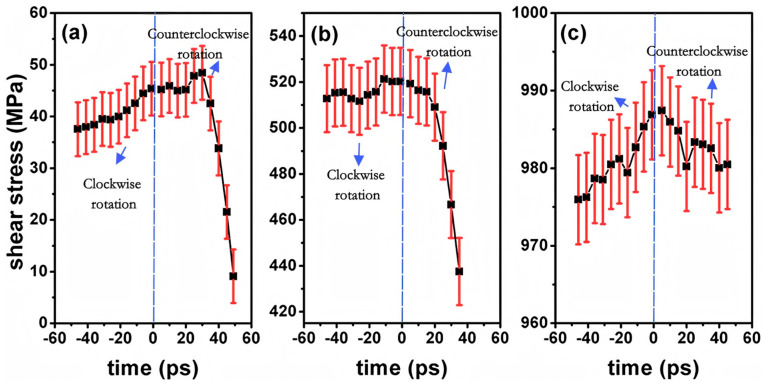
(**a**–**c**) are the shear stress curves of the GBs in Al, Ni, and Al/Ni at the top of the specimens in the [100] orientation [81].

**Figure 26 materials-18-03048-f026:**
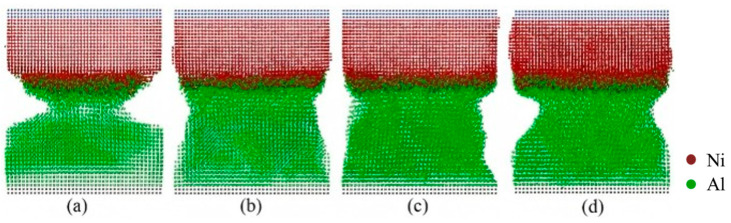
Planar view of diffusion-bonded Ni/Al bimetals after tension: (**a**) 1 K, (**b**) 300 K, (**c**) 500 K, and (**d**) 700 K [72].

**Figure 27 materials-18-03048-f027:**
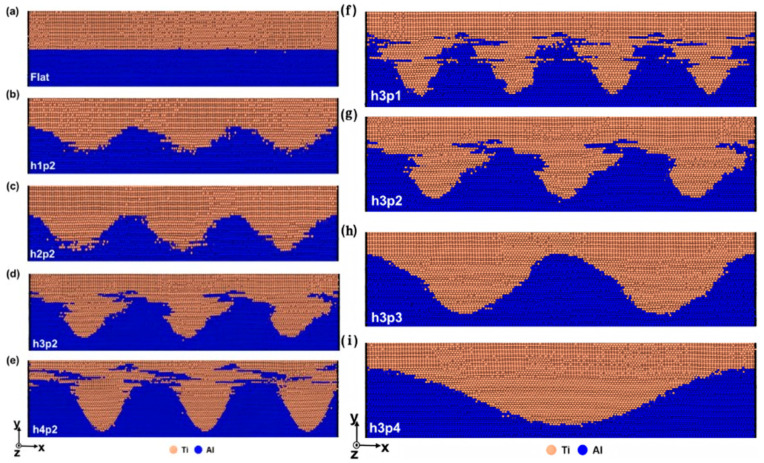
Atomic structure of planar and eight sinusoidal interface models after vibrational loading: (**a**) planar model, (**b**) h1p2, (**c**) h2p2, (**d**) h3p2, (**e**) h4p2, (**f**) h3p1, (**g**) h3p2, (**h**) h3p3, and (**i**) h3p4 models [83].

**Figure 28 materials-18-03048-f028:**
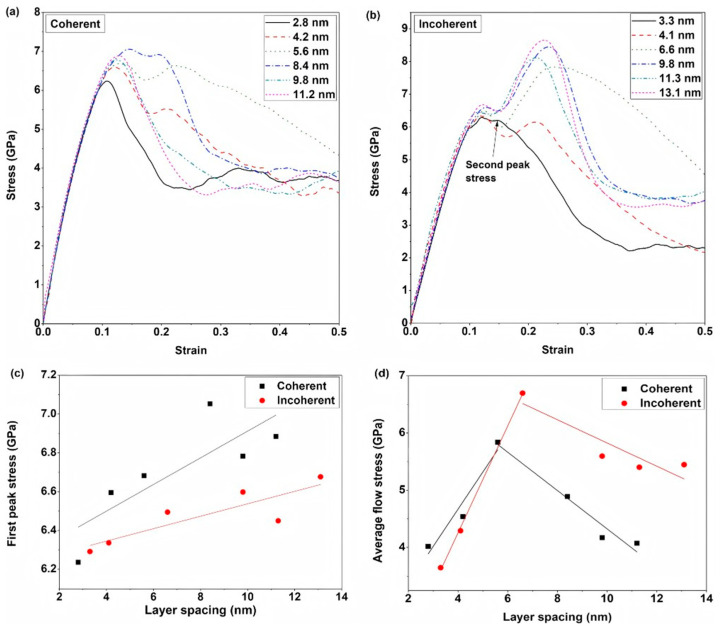
Stress–strain curves of Ti/Al nanolaminates with (**a**) the coherent interface and **(b**) the incoherent interface as layer spacing increases under tension loading; the variation of (**c**) the first peak stress and (**d**) the average flow stress with layer spacing [90].

**Figure 29 materials-18-03048-f029:**
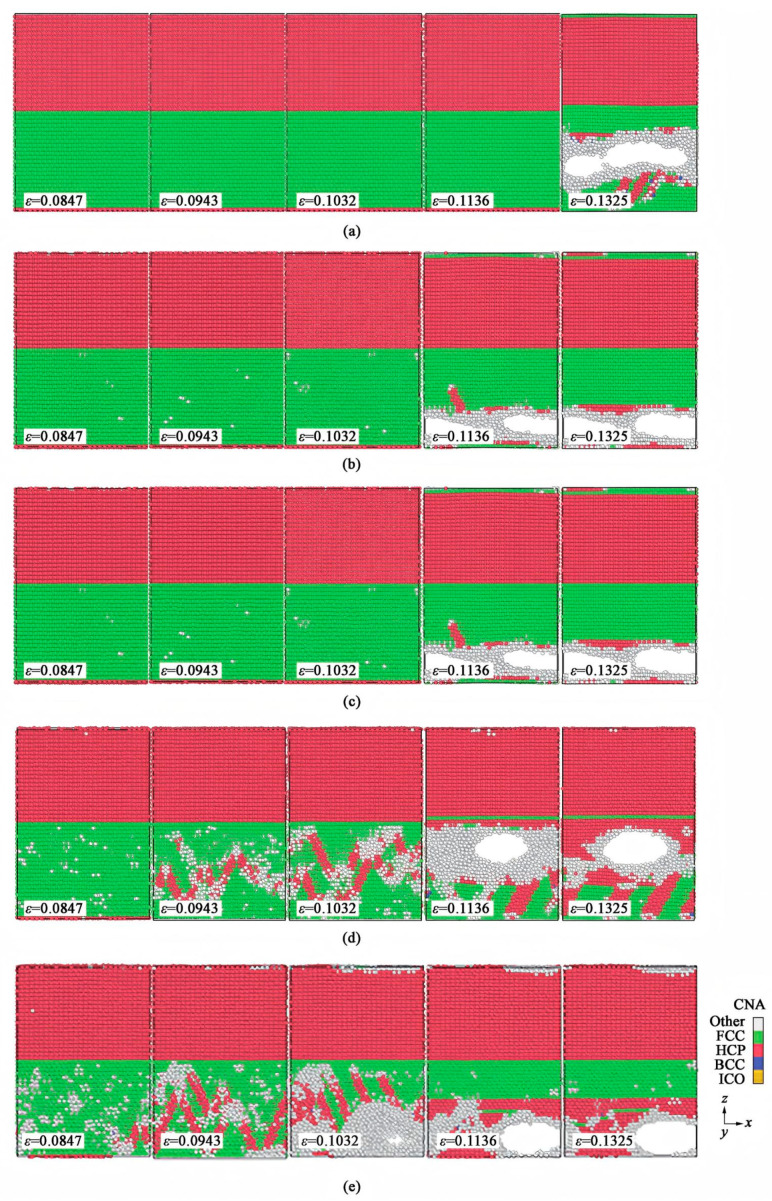
The atomic structure evolution of the Al/Ti interface model stretched at different cooling temperatures: (**a**) 100 K, (**b**) 200 K, (**c**) 300 K, (**d**) 400 K, and (**e**) 500 K [86].

**Figure 30 materials-18-03048-f030:**
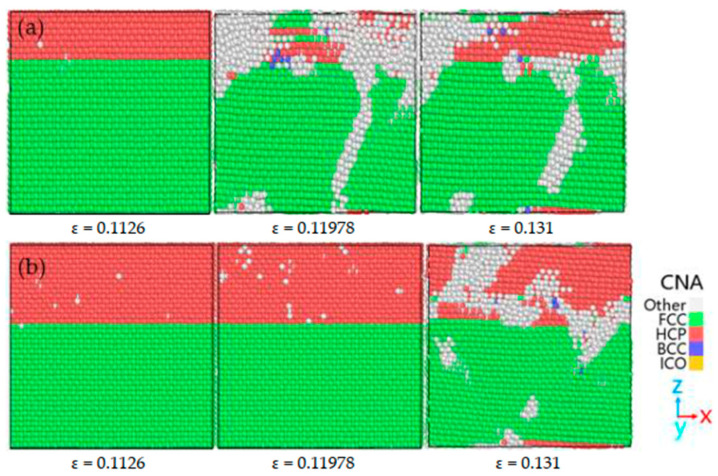
A simulation of the atomic structure evolution of Ti/Al with different temperatures and different Ti volume fractions: (**a**) 100 K, (**b**) 600 K [93].

**Figure 31 materials-18-03048-f031:**
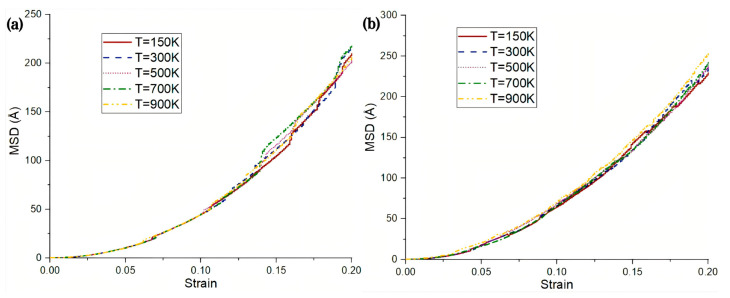
The mean squared displacement of different atoms as a function of strain from 150 K to 900 K: (**a**) Fe in Al and (**b**) Al in Fe [94].

**Figure 32 materials-18-03048-f032:**
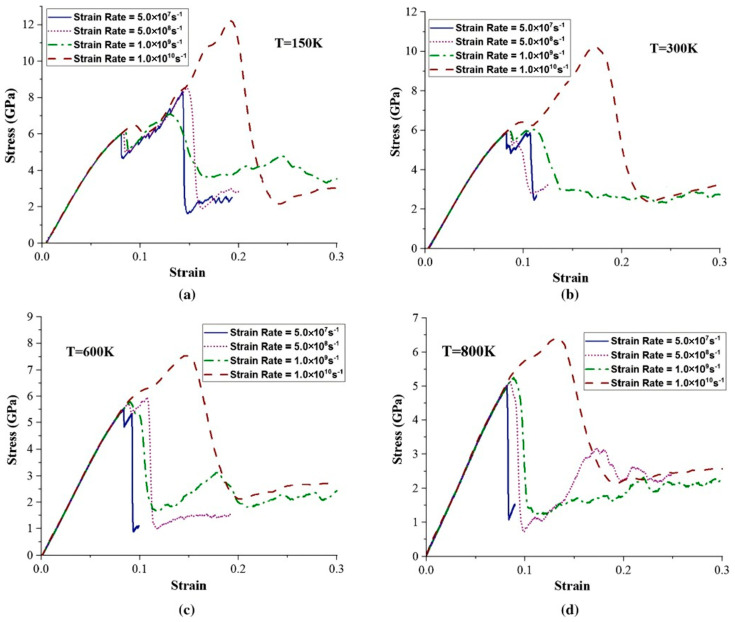
Al/Fe system stress versus strain curves: (**a**) T = 150 K, (**b**) T = 300 K, (**c**) T = 600 K, and (**d**) T = 800 K [97].

**Figure 33 materials-18-03048-f033:**
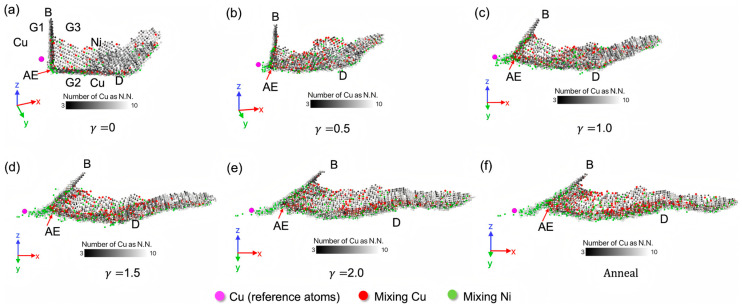
Sequential snapshots of the deformed cell at strain levels of (**a**–**e**) 0 to 2.0 and (**f**) annealing stage [98].

**Figure 34 materials-18-03048-f034:**
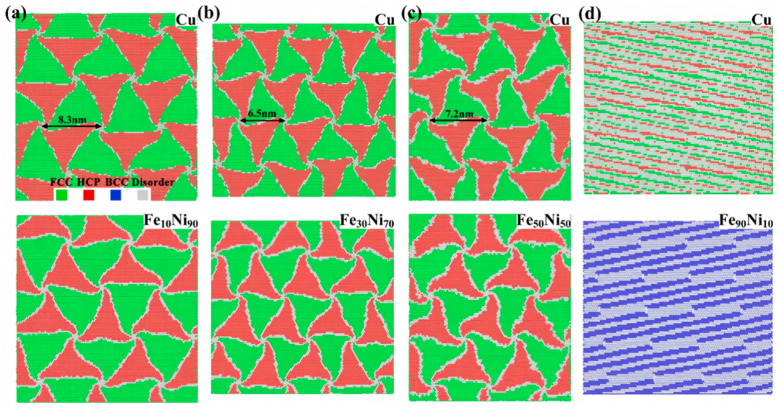
Interface structures of Cu/Fe_x_Ni_1−x_ layered composites: (**a**) Cu/Fe_10_Ni_90_, (**b**) Cu/Fe_30_Ni_70_, (**c**) Cu/Fe_50_Ni_50_, and (**d**) Cu/Fe_90_Ni_10_ [100].

**Figure 35 materials-18-03048-f035:**
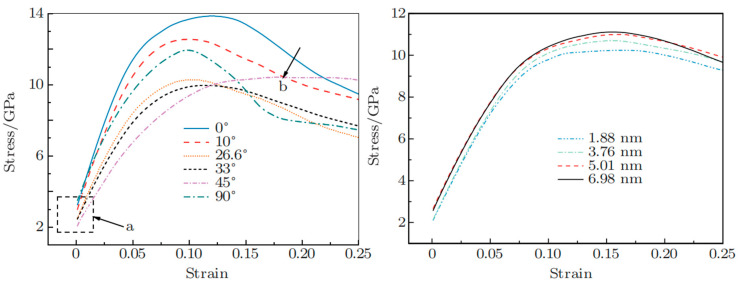
Stress–strain curves for the Cu/Ni: (**a**) different angles and (**b**) the 45°samples with different layer thickness values [104].

**Figure 36 materials-18-03048-f036:**
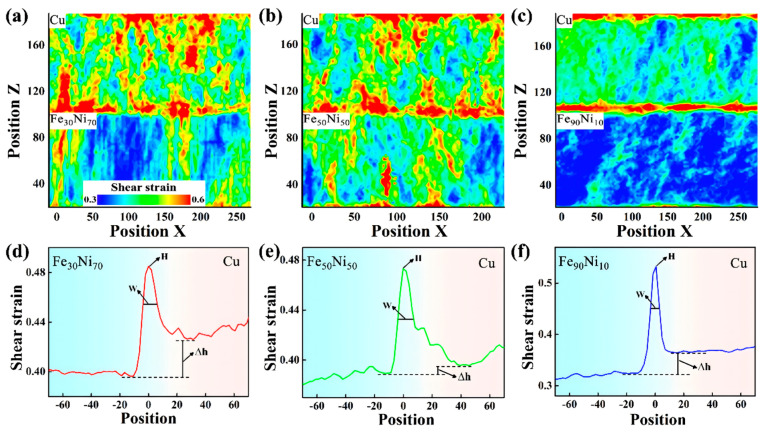
Shear strain distribution of Cu/Fe_x_Ni_1−x_ layered composites: (**a**–**c**) planar views and (**d**–**f**) views along the interface thickness direction [100].

**Figure 37 materials-18-03048-f037:**
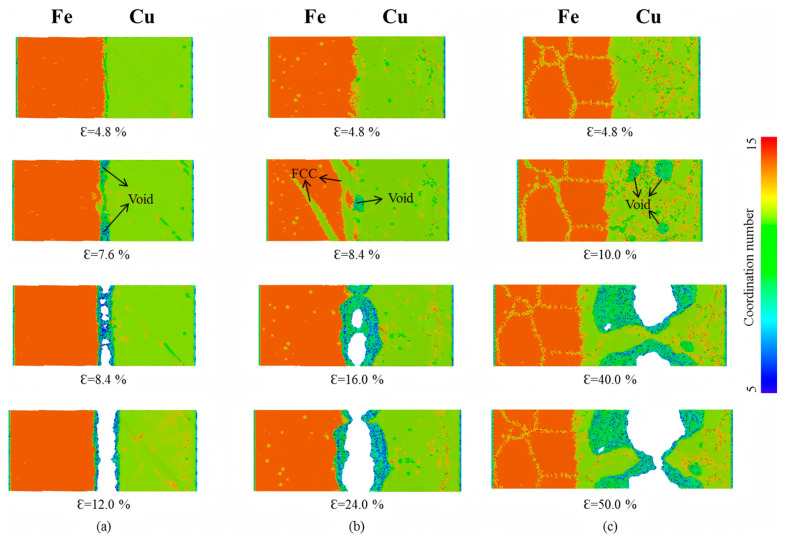
The deformed configurations of the bonding interfaces at different strains, when (**a**) Vz = 1500 m/s, (**b**) Vz = 1750 m/s, and (**c**) Vz = 2000 m/s [113].

**Figure 38 materials-18-03048-f038:**
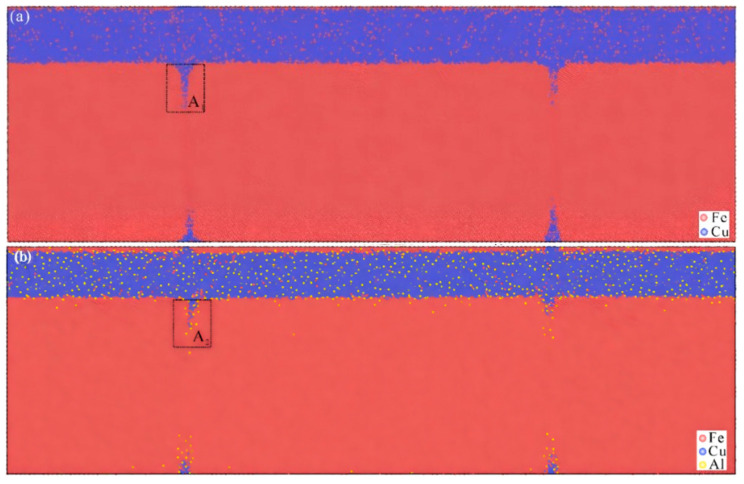
Snapshot of different systems in the liquid state at 20 ns: (**a**) liquid Cu penetration in Fe bicrystal and (**b**) liquid Cu(Al) penetration in Fe bicrystal [116].

**Table 1 materials-18-03048-t001:** The diffusion coefficient of Cu and Al at different temperatures.

Researcher	Temperature/K	Diffusion Coefficient of Al/(m^2^·s^−1^)	Diffusion Coefficient of Cu/(m^2^·s^−1^)	Atoms Conditions
Han XJ et al. [42]	650	9.64471 × 10^−12^	2.19764 × 10^−12^	Al (solid) Cu (solid) diffusion bonding
	700	2.78636 × 10^−11^	5.00352 × 10^−12^	
	750	6.24829 × 10^−9^	5.05976 × 10^−10^	
	800	6.85999 × 10^−9^	1.04891 × 10^−9^	
Cui YF et al. [43]	873	3.08 × 10^−12^	1.73 × 10^−9^	Al (solid → liquid) Cu (solid) Existence of IMCs
	893	8.05 × 10^−12^	1.86 × 10^−9^	
	913	4.49 × 10^−11^	1.94 × 10^−9^	
	933	2.68 × 10^−10^	2.05 × 10^−9^	
	953	5.15 × 10^−10^	2.17 × 10^−9^	
Mao AX et al. [44]	800	1.3140 × 10^−9^	2.12 × 10^−11^	Al (liquid) Cu (solid)
	850	1.5785 × 10^−9^	1.004 × 10^−10^	
	900	1.7770 × 10^−9^	2.434 × 10^−10^	
	950	2.0755 × 10^−9^	4.833 × 10^−10^	
Qian XF et al. [45]	953	6.3017 × 10^−9^	9.475465 × 10^−9^	Al (liquid) Cu (solid)
	973	7.28007 × 10^−9^	1.0293025 × 10^−8^	
	993	8.36446 × 10^−9^	1.0813525 × 10^−8^	
	1013	9.388485 × 10^−9^	1.1257745 × 10^−8^	
	1033	1.0975015 × 10^−8^	1.1947165 × 10^−8^	

**Table 2 materials-18-03048-t002:** Lattice constant and in-plane mismatch for all models of Cu/Al multilayer.

Models	Crystal Orientation Along x-Axis	Crystal orientation Along the y-Axis	Crystal orientation Along the z-Axis	Lattice Mismatch	In-Plane Mismatch Along x-Axis	In-Plane Mismatch Along y-Axis
(111)-A	[$\overset{-}{1}10$]	[$\overset{-}{1}\overset{-}{1}2$]	[111]	11.35%	15.38	8.879
(001)-B	[100]	[010]	[001]	11.35%	21.75	21.75
($\overset{-}{1}10$)-C	[$\overset{-}{1}\overset{-}{1}2$]	[111]	[$\overset{-}{1}10$]	11.35%	8.879	25.11
($\overset{-}{1}\overset{-}{1}2$)-D	[111]	[$\overset{-}{1}10$]	[$\overset{-}{1}\overset{-}{1}2$]	11.35%	25.11	15.38

**Table 3 materials-18-03048-t003:** The Young’s moduli and yielding strengths of the homogenous and heterogeneous Al/Cu bimetals with ($\overset{-}{1}\overset{-}{1}1$) and (001) interfaces.

Mechanical Properties	Homogenous Sample with ($\overset{-}{1}\overset{-}{1}1$) **Interface**	Heterogenous Sample with ($\overset{-}{1}\overset{-}{1}1$) **Interface**	Homogenous Sample with (001) Interface	Heterogenous Sample with (001) Interface
Young’s moduli	127.6 GPa	129.3 GPa	33.8 GPa	31.9 GPa
Yielding strengths	9.5 GPa	8.4 GPa	1.5 GPa	1.3 GPa

**Table 4 materials-18-03048-t004:** Diffusion coefficients of Al and Ti atoms at 973 K.

Diffusion Coefficients	Ideal Interface	Rough Interface
D_Al_ (m^2^/s)	2.6667 × 10^−9^	2.4333 × 10^−9^
D_Ti_ (m^2^/s)	1.65 × 10^−11^	1.47 × 10^−11^

**Table 5 materials-18-03048-t005:** Tensile strength of Cu($\overset{-}{1}1\overset{-}{1}$)/Fe(110)-KS and Cu($\overset{-}{1}1\overset{-}{1}$)/Fe(110)-NW interfaces, as well as that of pure Cu and Fe in the x-, y-, and z-directions.

Tensile Direction	Interface or Pure Metal	Tensile Strength (GPa)
X (Parallel to the interface)	Cu($\overset{-}{1}1\overset{-}{1}$)/Fe(110)-KS	9.68
	Cu($\overset{-}{1}1\overset{-}{1}$)/Fe(110)-NW	6.64
	pure Cu along [011]	6.28
	pure Fe along [$\overset{-}{1}11$]	35.64
	pure Fe along [001]	9.90
Y (Parallel to the interface)	Cu($\overset{-}{1}1\overset{-}{1}$)/Fe(110)-KS	11.08
	Cu($\overset{-}{1}1\overset{-}{1}$)/Fe(110)-NW	11.25
	pure Cu along [$2\overset{-}{1}1$]	10.74
	pure Fe along [$1\overset{-}{1}2$]	22.30
	pure Fe along [$1\overset{-}{1}0$]	26.82
Z (Vertical to the interface)	Cu($\overset{-}{1}1\overset{-}{1}$)/Fe(110)-KS	10.25
	Cu($\overset{-}{1}1\overset{-}{1}$)/Fe(110)-NW	11.99
	pure Cu along [$\overset{-}{1}1\overset{-}{1}$]	13.08
	pure Fe along [110]	26.82

**Table 6 materials-18-03048-t006:** Comparison of diffusion coefficients in different bimetallic systems.

Different Systems	Diffusion Coefficient (m^2^·s^−1^)		Simulated Condition
	Al or Fe	Cu or Mg or Ti	
Al/Mg [62]	278 × 10^−11^	451 × 10^−11^	β = 10°
	334 × 10^−11^	456 × 10^−11^	V_p_ = 440 m/s
Al/Ni [124]	0.43	0.27	0 V/Å
	0.46	0.31	5 × 10^−11^ V/Å
	0.56	0.36	1 × 10^−10^ V/Å
	0.92	0.58	2 × 10^−10^ V/Å
Al/Ti [84,86,87]	266.67 × 10^−11^	1.65 × 10^−11^	ideal interface
	243.33 × 10^−11^	1.47 × 10^−11^	rough interface
	35.5 × 10^−11^	27.3 × 10^−11^	600 K and 50 MPa
	19.2 × 10^−11^	329.0 × 10^−11^	1340 K
Fe/Cu [108,110]	0.644 × 10^−11^	303.1 × 10^−11^	1373 K
	1.227 × 10^−11^	336.4 × 10^−11^	1423 K
	2.711 × 10^−11^	368.2 × 10^−11^	1473 K
	4.806 × 10^−11^	394.8 × 10^−11^	1523 K
	10.084 × 10^−11^	446.4 × 10^−11^	1573 K
	0.353 × 10^−11^	491.7 × 10^−11^	(100)
	0.464 × 10^−11^	549.6 × 10^−11^	(110)
	0.554 × 10^−11^	512.6 × 10^−11^	(111)

**Table 7 materials-18-03048-t007:** The research directions and progress of bimetals in recent years.

Types	Research Contents
Al/Cu	Atomic diffusion behavior [39,40,42,45,46,57]
	Coefficient of diffusion [42,44,45]
	Diffusion depth [44]
	Rate of diffusion [44]
	Activation energy of diffusion [57]
	Diffusion mechanism [57]
	Transitional layer thickness [41,42,44,45,47,57]
	Plastic deformation mechanism [39,50,52,54,56,57,58]
	Tensile strength [41,42,45,51,52,55,56,57,58,59]
	Yield strength [42,45,46,58]
	Young’s modulus [59]
	Broken mode [51,52,55,56]
	Shear strength [55]
	Interface reinforcement mechanism [53]
	The solidification process [40]
Al/Mg	Atomic diffusion behavior [62,67]
	Coefficient of diffusion [62]
	Diffusion rate [62]
	Diffusion layer thickness [67]
	Modulus of elasticity [64]
	Plastic deformation mechanism [64]
	Tensile property [63]
	Shear strength [67]
	Breaking mechanism or mode [63,66,67]
	Interface reinforcement mechanism [64]
Al/Ni	Atomic diffusion behavior [71,73,74,75]
	Coefficient of diffusion [75]
	Diffusion rate [73,74]
	Diffusion mechanism [75,80]
	Diffusion layer thickness [72,73,74]
	Intermetallic compound formation species [76,78]
	Formation mechanism of intermetallic compounds [77]
	Tensile property [72]
	Plastic deformation mechanism [72]
	Interface dislocation behavior [81]
	Evolution behavior of crystal boundary [81]
Al/Ti	Dispersal behavior [85]
	Diffusion constant [84,86]
	Formation mechanism of intermetallic compounds [87,88,89]
	Tensile property [85,86]
	Broken mode [84,86]
	Plastic deformation mechanism [86,90,92]
	Structural evolution [90,92]
	Compression performance [93]
Al/Fe	Atomic diffusion behavior [94,96]
	Studies on intermetallic compound generation [94]
	Formation mechanism of intermetallic compounds [95]
	Anti-shear performance [96]
Cu/Ni	Diffusion mechanism [98]
	Transitional layer thickness [99]
	Tensile property [99,104,106,107]
	Anti-shear performance [106]
	Plastic deformation mechanism [98,100,101,106]
	Interface reinforcement mechanism [101,102]
	Structural evolution [98,105]
Fe/Cu	Coefficient of diffusion [108,109,110]
	Diffusion depth [108]
	Diffusion rate [116]
	Diffusion mechanism [108]
	Diffusion length [108,109,110]
	Tensile property [110]
	Plastic deformation mechanism [115]
	Connectionism [113]
	Interface reinforcement mechanism [109]
	Combined interface type [113]

## Data Availability

The authors confirm that the data supporting the findings of this study are available within the article.

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
