# Peer review of "Review of Molecular Dynamics Simulation of Bimetallic Interfacial Behavior"

_materials, 2025, doi:10.3390/ma18133048_

Round 1

Reviewer 1 Report

Comments and Suggestions for Authors

This review systematically summarizes the progress in molecular dynamics (MD) simulations of bimetallic interfaces. The manuscript is well written and orgnized, however, further improvements are needed in the following aspects before final decision:

  • The comparative analysis of external factors (such as temperature) across different systems lacks systematic organization. It is recommended to add a "Key Factor Comparison" table to summarize data such as diffusion coefficients and phase transition temperatures, and annotate the reasons for differences between systems.
  • Some charts have incomplete axis labels and superficial data interpretation. It is necessary to complete the labels (e.g., add units) and perform Arrhenius fitting on diffusion coefficients to calculate activation energy.
  • There is insufficient citation of cutting-edge technologies (such as machine learning potential functions challenges) from the recent three years (2023-2025). It is recommended to supplement relevant literature on "AI-driven MD simulation" and discuss its applications in interface simulation.
  • There is insufficient citation of Bimetallic Interface Behavior in the recent three years. It is recommended to supplement with the following relevant literature:

https://doi.org/10.1016/j.carbon.2025.120109

https://doi.org/10.1016/j.compositesa.2025.108741

  • The future outlook needs specific technical roadmaps. For example, use the Al/Ti system as a case to illustrate the integration of MD and finite element methods, or cite simulation cases where ultrasonic vibration (e.g., amplitude of 5 nm) regulates interface diffusion.

Reviewer 2 Report

Comments and Suggestions for Authors

The manuscript entitled with “  Review about Molecular Dynamics Simulation of Bimetallic  Interface Behavior   ” is a research paper and brings some relevant information about the application of bimetals in many fields due to the combination of the performance characteristics of the two materials. Still, the weak interface bonding limits their promotion and application. So, . Molecular dynamics (MD) is an advanced microscopic technique that reveals the nature of interface bonding and the mechanism of strengthening.

It is an original study, but the following considerations will improve it:

  1. In the Introduction Section, line 87, about Figure 1, it has much information that the author does not describe.
  2. About Figure 1, where are the references?
  3. For all bimetallic Molecular Dynamics Simulation Interface Behavior Sections, there are some problems with subtitles. Sometimes the authors use markers, sometimes use subtitles.

4.. In the Molecular Dynamics Simulation Interface Behavior Section, Figure 2-11 has a problem with its caption.

  1. The author must include an overall topic of Molecular Dynamics Simulation using to characterize the bimetallic materials behavior, using actualized references.
  2. In the conclusion section, the authors need to describe how all the properties of Molecular Dynamics Simulation investigated and discussed with different bimetallic materials will be useful for daily applications.

Reviewer 3 Report

Comments and Suggestions for Authors

Study of bimetallic interfaces is of great interest because the interaction between the metals can significantly alter their properties, including their electronic behavior, mechanical strength, and reactivity. Authors performed valuable research and analyzed recent data on molecular dynamics concerning the behavior of bimetallic interfaces, the most attention is paid to the interface formation process and mechanical behavior. Seven bimetallic systems are considered; internal and external factors are analyzed. The material is well presented and perfectly illustrated. Future prospect in the corresponding scientific direction is highlighted. The given data on compression behavior, molecular dynamics simulation and other points are interesting for a wide audience.

Main question addressed by the research is summarizing data on interfacial behavior of Al/Cu, Al/Mg, Al/Ni, Al/Ti, Al/Fe, Cu/Ni, and Fe/Cu bimetals, the scope of review lies within interfacial atomic diffusion behavior and their properties, diffusion coefficients, depths, and mechanisms. The topic is original, because molecular dynamics represents convenient, rapid and accurate study of mechanical behavior at the bimetallic interfaces.

The specific gap in the field of research is covered, due to the study is focused on the microscopic mechanisms of the thickness and formation of intermetallic compounds, and the structural evolution process. Sufficient data is added to the subject area compared with other published material: authors explored future research directions in molecular dynamics, offered theoretical guidance for understanding of the interface behavior mechanisms and performance optimization of bimetallic materials.

The topic is presented quite fully. No improvements are required by the addition of material, and demonstrating something else in the methodology.

The conclusions are consistent with the evidence and arguments presented, they address the main question properly, because plastic deformation mechanisms, tensile properties, fracture modes and interfacial strengthening mechanisms were also chosen as the focus of the studies.

The references are totally appropriate, the formatting of the list of references is perfect.

Additional comments on the figures: Please check if some figures need the notice “Reproduced with permission of (some publishing house)”.

The manuscript can be published after minor revision. Please also correct the following points:

Line 24 – “bimetallic” instead of “bemetallic”

Line 944 – 6 summarizes instead of “6summarizes”

Reviewer 4 Report

Comments and Suggestions for Authors

The Authors presented a study regarding the studying the interface behavior to achieve the bimetallic strengthening. The research presented is interesting and in my opinion the article deserves to publish in Materials after a minor correction. I would like to suggest introducing changes before publishing in Materials.

Only points out some of the article's weak points:

  1. Double check all shortcuts. They must be entered before they can be used.
  2. Figure 2-1: The article uses two different numberings of compounds. Please unify the numbering system by using consecutive numbers.
  3. Table 1: Please remove the temperature range column.
  4. The conclusions are not sufficient. Please supplement them.
  5. The article contains many typos. Please check the article again for this.
Comments on the Quality of English Language

The language requires a lot of corrections. There are a lot of errors
